# UNLEARNING PARADOX: AUDITING RESIDUAL IDENTITY TRACES IN FACE RECOGNITION

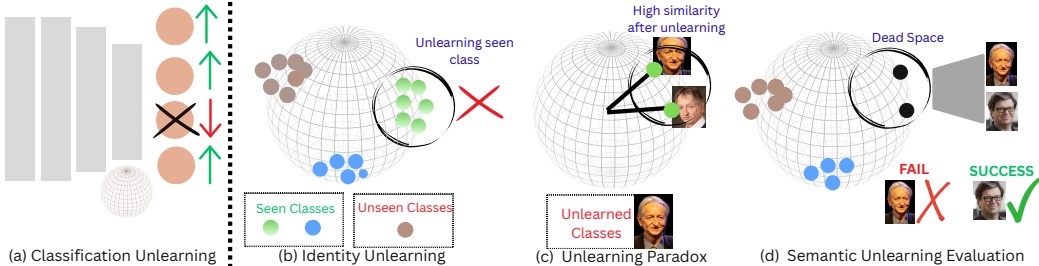

Figure 1: Illustration of the unlearning paradox in face recognition. (a) Traditional classification unlearning evaluates success by reduced accuracy on forgotten classes in a closed-set scenario. (b) In face recognition, embeddings enable verification of both seen and unseen identities, making simple class removal inefficient. (c) Even after unlearning, the samples of forgotten identity exhibit high similarity, revealing the paradox. (d) We propose semantic unlearning evaluation, where success is determined by the absence of recoverable identity semantics rather than accuracy degradation.

## ABSTRACT

Face recognition systems raise a critical privacy question: how do we prove that a person's biometric data has been deleted when laws such as GDPR or CCPA require it? We highlight an unlearning paradox - A model can still verify "forgotten" identities because face recognition works in an open set, where unseen identities remain recognizable. This makes standard accuracy-based tests misleading. We contribute three ideas. (1) We formalize this paradox and show why current metrics give a false sense of forgetting. (2) We design a generative auditing framework that reconstructs faces from embeddings, exposing that existing methods keep up to 57% of identity information even when they appear to succeed. (3) We propose FUSE (Forgetting Using Structural Erasure), which treats identities as hypercones and erases them with region-aware surrogates while preserving recognition of others. On CASIA-WebFace and D-LORD, FUSE reduce the amount of semantic residual ($>0.6$) for forget set while retaining high verification for non-target classes. Our work shifts evaluation from accuracy to semantics, setting stronger privacy standards for face recognition.

## 1 INTRODUCTION

When a European citizen discovers their face in a commercial recognition system and invokes their "right to be forgotten" under GDPR Article 17, the system must delete their biometric data. But here's the problem: how can we verify that their identity has actually been erased from the model, not just the database? With over 1 billion faces enrolled in recognition systems worldwide Jain et al. (2024), this is not a hypothetical concern, rather it is a daily privacy challenge with legal ramifications.

Current machine unlearning methods fail catastrophically when applied to face recognition Nguyen et al. (2022). These methods, designed for classification tasks Li et al. (2025), assume that forgetting success can be measured by accuracy degradation on removed classes. But face recognition operates fundamentally differently Deng et al. (2019). These systems work by computing embedding similarities in an open-set scenario, where the model must recognize identities it has never seen during

training Günther & Boult (2017). This leads to what we call the **unlearning paradox**: if a face recognition model truly maintains its discriminative power, it should successfully verify a "forgotten" identity just as it verifies any unseen person. The very property that makes face recognition useful by generalizing to unseen identities, which makes it impossible to verify forgetting through conventional metrics Sekhari et al. (2021).

This paradox is not just theoretical. When we audit existing unlearning methods through generative reconstruction, we find they retain up to 57% of identity information in the embedding space, even when verification accuracy appears to drop. Methods like gradient ascent Thudi et al. (2022), boundary shifting Chen et al. (2023), and saliency-based unlearning Fan et al. (2024) all suffer from this fundamental issue. The forgotten faces can still be reconstructed from their supposedly erased embeddings. Privacy regulations require verifiable deletion, but current evaluation protocols using match/non-match tests Carlini et al. (2022) fundamentally cannot distinguish between a truly forgotten identity and one that simply behaves like an unseen person.

We need a complete rethinking of how unlearning works in embedding spaces. Rather than trying to degrade verification accuracy, which contradicts the model's purpose, we must ensure that identity-specific semantic information cannot be recovered from the feature space. This requires understanding how identities are geometrically encoded Wang et al. (2018); Liu et al. (2017) and developing methods to surgically remove these representations while preserving the overall discriminative structure Wang et al. (2025).

In this paper, we formalize the unlearning paradox and demonstrate why conventional evaluation fails for open-set recognition systems (refer to Figure 1 for visual understanding). We introduce semantic residual auditing as a stronger evaluation framework that quantifies hidden identity leakage directly from the embedding space, without requiring image generation. Based on these insights, we propose FUSE (Forgetting Using Structural Erasure), a geometry-aware method that models identity distributions as hypercones and performs targeted semantic erasure. Unlike existing approaches that modify classifiers Tarun et al. (2023) or individual embeddings, FUSE transforms the geometric structure of identity regions, achieving demonstrable privacy protection while maintaining recognition capability.

Our contributions are:

- **Theoretical Foundation:** We formalize the unlearning paradox, proving that verification-based evaluation cannot distinguish between successful forgetting and natural generalization in open-set systems Sekhari et al. (2021); Günther & Boult (2017).

- **Generative Auditing Framework:** We propose a reconstruction-free evaluation technique, termed Semantic Residual, that reveals existing methods Fan et al. (2024); Chen et al. (2023) leak substantial identity information despite apparent forgetting.

- **FUSE Method:** We introduce a geometry-aware unlearning approach using hypercone modeling and region-aware surrogate representations that achieves near-zero semantic leakage, building on geometric insights from Deng et al. (2019); Wang et al. (2018).

- **Comprehensive Evaluation:** We demonstrate across multiple datasets that FUSE eliminates identity reconstruction while preserving 85% discriminability, setting new standards for privacy-compliant face recognition.

## 1.1 RELATED WORK

**Classification based unlearning:** Most existing unlearning techniques focus primarily on the classifier layer Fan et al. (2024); Zhou et al. (2025), often by removing or fine-tuning the weights associated with specific output logits or applying parameter resets Tarun et al. (2023); Kodge et al. (2024). Unlearning in such cases can be performed via gradient ascent Golatkar et al. (2020); Cha et al. (2024); Thudi et al. (2022) strategy for the data to be forgotten while using gradient descent for the retaining data. Differential privacy-based techniques Gupta et al. (2021) were introduced in machine unlearning to limit the retention of sensitive information during retraining. Projection-based unlearning methods remove information from the parameter or embedding space by projecting gradients onto the null space of the forgotten data Chen et al. (2024); Hoang et al. (2024); Ilharco et al. (2023).

More recently, adversarial strategies have been explored Shan et al. (2020); Di et al. (2025), where unlearning is achieved by generating adversarial perturbations. Another approach involves boundary shifting Chen et al. (2023), which shrinks the decision space of the forgetting class and expands the boundary of a shadow class to remove targeted information. Apart from sample unlearning, Hayase et al. (2020) formalized unlearning at finer levels to forget selective information by framing forgetting as an optimization problem that balances removal and retention.

**Feature Level unlearning:** Unlike traditional unlearning methods that focus on classification models, several approaches have been formulated for feature removal. These methods focus on shifting features using contrastive loss Wang et al. (2025) and alignment Wang et al. (2024). Supervision-free Shen et al. (2024) unlearning models target the cluster as class labels are absent. Influence functions Warnecke et al. (2021) are used to approximate the effect of training data on model parameters and update using first and second-order derivatives rather than doing iterative optimization-based retraining. Meanwhile, in contrastive learned models like SimCLR Chen et al. (2020), MoCo He et al. (2020) and CLIP Radford et al. (2021), unlearning is performed by targeting the unique characteristics of contrastive learning. Wang et al. (2025) modifies the contrastive learning objective through alignment calibration by incorporating a positive term to reduce sensitivity to unlearned data.

Unlearning in face recognition models in the literature is limited to classification settings or attribute-level unlearning Choi & Na (2023); Guo et al. (2022). It progressively detaches selective features from the learned model by learning correlations in the latent space and each feature's effect on the output space. Our work fundamentally differs from existing approaches by targeting the geometric structure of the feature space for the data to be forgotten, focusing on the overall class distribution rather than individual logits or sample-level features.

## 2  RETHINKING IDENTITY UNLEARNING IN EXISTING SYSTEMS

In face recognition, the goal is to check if a query image $x_q$ matches any enrolled identity from a gallery $\{x_g^1, x_g^2, \ldots, x_g^n\}$. Unlike closed-set classification, these systems work in an open-set setting. Instead of predicting class labels, the model uses a feature extractor $f_\theta : \mathbb{R}^{H \times W \times C} \to \mathbb{R}^d$ to map both query and gallery images into a $d$-dimensional embedding space. For two images, verification using feature space is done by comparing embeddings with a similarity metric such as cosine similarity:

$$\text{sim}(f_\theta(x_q), f_\theta(x_g^i)) \geq \tau$$

If the similarity exceeds a threshold $\tau$, the system declares a match; otherwise, it declares a mismatch. This process relies only on the geometry of the embedding space and ignores the classifier head used in training.

Formally, let the training set be $D_{tr} = \{(x_i, y_i)\}_{i=1}^N$, where $x_i$ are face images and $y_i \in Y$ are identity labels. The model is trained with a discriminative loss $\mathcal{L}_{disc}$ (e.g., ArcFace Deng et al. (2019), CosFace Wang et al. (2018)) so that embeddings of identities in $Y$ are well separated. At test time, the system sees a new set $D_{te} = \{x_j'\}_{j=1}^M$ with identities $z_j \in Z$, where $Z$ may be disjoint or only partially overlap with $Y$ ($Z \cap Y = \varnothing$). Despite being trained on $Y$, the model must generalize to unseen identities in $Z$ by encoding features that remain discriminative Sekhari et al. (2021). Verification at test time therefore depends only on embedding similarity, not on class predictions.

Now consider unlearning. Suppose we want to forget a subset of training identities $Y_u \subset Y$, with corresponding data $D_u = \{(x, y) \mid y \in Y_u\}$. After unlearning, we obtain an updated extractor $f_{\theta_u}$. In principle, the embeddings of $Y_u$ should no longer be tied to their original labels. In practice, however, the system still works in an open-set setting. Forgotten samples may behave like unseen identities: they still cluster together in the embedding space and remain verifiable under $f_{\theta_u}$. This means that even if the classifier layer forgets $Y_u$, the embeddings may still carry strong identity representations in the feature space.

We call this the **Unlearning Paradox**. Consider two cases:

- **Unseen identity:** If $Z$ was never part of training, the model can still verify it. This reflects generalization of the embedding space.

- **Forgotten identity:** If $Y_u$ was trained but later unlearned, we expect it to behave indistinguishably from an unseen identity, yet in practice, the embeddings of $Y_u$ may still form a cluster, allowing verification just as in the unseen case.

**Definition (Unlearning Paradox).** Let $Y$ be the set of training identities, $Y_u \subseteq Y$ the forget set, and $Z$ a set of test identities with $Z \cap Y = \varnothing$ (open set). Let $f_\theta$ be the encoder before, and $f_{\theta^-}$ after unlearning. With the verification decision $\mathrm{ver}_\tau(x, x'; f) = \mathbf{1}[\langle f(x), f(x')\rangle \geq \tau]$ and the same-identity verification rate $V(f, \tau; S)$ over pairs drawn from identity set $S$, the *unlearning paradox* occurs when

$$V\big(f_{\theta^-}, \tau; Y_u\big) \approx V\big(f_{\theta^-}, \tau; Z\big).$$

Intuitively: after unlearning, forgotten identities behave like unseen identities under verification, so closed-set accuracy drops or pass/fail verification on $Y_u$ alone cannot certify forgetting in open-set face recognition.

This paradox shows a contradiction. In open-set systems, verifying unseen identities is considered a success, but verifying forgotten ones is treated as a failure. However, both outcomes stem from the same embedding geometry. Verification measures discriminability, not label memory. Therefore, accuracy drops alone cannot prove that an identity has been erased. True forgetting in feature space must be assessed with additional probes, such as generative reconstruction or distributional shift analysis.

## 3 METHODOLOGY: FORGETTING USING STRUCTURAL ERASURE

In representation-based unlearning, the goal is to directly target and remove the latent representation of a class from the embedding space, rather than merely modifying classifier weights. The key idea is to ensure that even if test samples of a forgotten identity still cluster together for verification, but at different positions in the feature space, the model no longer retains the semantic traces that encode that identity. To achieve this, we propose FUSE (Forgetting Using Structural Erasure), which constructs surrogate representations from external samples, searches for their alignment with the target identity's representation in the learned feature space, and then erases these directions to suppress residual identity information.

### 3.1 SURROGATE REPRESENTATION RETRIEVAL

To erase the semantic footprint of an identity from the feature space, it is crucial to first characterize how that identity is distributed in the embedding space. As in the pixel space, it is easy to visualize a person; however, the representation in the learned embedding space is less intuitive. For multiple images of an identity, the encoder produces embeddings $\{z_i = f_\theta(x_i) \mid x_i \in \mathcal{D}_{Y_u}\}$, which is not a single point but instead span a local region on the unit hypersphere $\mathbb{S}^{d-1}$. This region can be understood as the geometrical support of the class within the embedding space, and any effective unlearning strategy must identify and remove the structural directions that define this support.

However, a face is composed of semantically distinct subregions (such as the nose, lips, and eyes), each contributing differently to the embedding. Simply unlearning from complete face images fails to capture the intra-class variation arising from these subregions. To expose these variations, we employ region-based embeddings in addition to whole-face embeddings. Let $z^{\mathrm{face}} \in \mathbb{R}^d$ denote a complete face embedding and $z_r^{\mathrm{reg}} \in \mathbb{R}^d$ the embedding of region $r \in \{\mathrm{nose}, \mathrm{eyes}, \mathrm{mouth}, \dots\}$. By stochastically combining these subregional embeddings, we synthesize surrogate representations that span the latent variability of the identity. This process is implemented using a vector database that contains embeddings of full images and regions obtained by performing a masking operation. For a target identity $y_u$, we retrieve: (i) its samples embeddings $\{z_i^{\mathrm{face}}\} \sim$ 5-10, (ii) its region embeddings $\{z_{r,i}^{\mathrm{reg}}\}$, (iii) its prototype embedding $\bar{z}_{y_u} = \frac{1}{N}\sum_i z_i$, and (iv) prototypes of semantically close neighbors $\{\bar{z}_{y_n}\}$ identified using cosine similarity. These retrieved vectors collectively serve as surrogate traces that approximate how the forgotten identity is encoded in the representation space, forming the foundation for structural erasure in subsequent steps.

## 3.2 STRUCTURAL TRACE IDENTIFICATION

A single facial region, such as the eyes or nose, is not unique enough to reliably represent an identity, since many individuals may share similar local features. However, when different regions are combined together, they form a unique signature that characterizes a face. In face recognition models trained with margin-based losses like ArcFace, each regional embedding lies on a hypersphere, where angular distances encode similarity. To ensure that combinations of regions also respect this spherical geometry, we generate fused embeddings using spherical linear interpolation (SLERP) rather than simple averaging. Given two normalized regional embeddings $z_{1,i}^{\text{reg}}$ and $z_{2,i}^{\text{reg}}$ for class $i$, we form a composite embedding as:

$$z_i^{\text{combo}} = SLERP(z_{1,i}^{\text{reg}}, z_{2,i}^{\text{reg}}, \alpha) = \frac{\sin((1-\alpha)\theta)}{\sin\theta} z_{1,i}^{\text{reg}} \frac{\sin(\alpha\theta)}{\sin\theta} z_{2,i}^{\text{reg}}$$

where $\theta = \arccos(z_{1,i}^{\text{reg}\top} z_{2,i}^{\text{reg}})$ is the geodesic angle between them and $\alpha$ is interpolation parameter. To learn which variations are truly identity-specific, we use a contrastive learning objective. A small projection network $p_\sigma : \mathbb{R}^d \to \mathbb{R}^d$ is used, which aims to map fused embeddings to the structural space of the class. Positive pairs are embeddings from different samples of the same identity, while negatives come from different identities. With normalized features $s_i = p_\sigma(z_i^{\text{combo}})$, the contrastive loss is:

$$\mathcal{L}_{\text{struct}} = -\sum_{i,j \in Y_u} \log \frac{\exp(s_i^\top s_j/\tau)}{\sum_{k \in Y} \exp(s_i^\top s_k/\tau)},$$

which encourages embeddings of the same person (under different regional combinations) to stay close, while separating them from other identities. The loss is also used to enforce the combined embedding with the class prototype using context-based dropout to introduce intra-class variation. With this, we obtain a set of candidate vectors $s_i$ that contains variations to represent the target class.

## 3.3 TARGET REGION FORMATION

In our proposed FUSE algorithm, we aim to obtain a "dead space" in the feature space region where a target class used to reside. With the obtained candidate features $s_i$, where $i = 1, 2, \ldots, k$ belonging to a class $Y_u$ (forget set), we aim to localize a region. Rather than feature-based unlearning, our proposed FUSE algorithm aims to focus on class-geometry-based unlearning. With our theoretical understanding of face recognition models Deng et al. (2019), samples of a class $Y_u$ follow angular relationships and form a hyperconical structure on the surface of a high-dimensional hypersphere as shown in Figure 2.

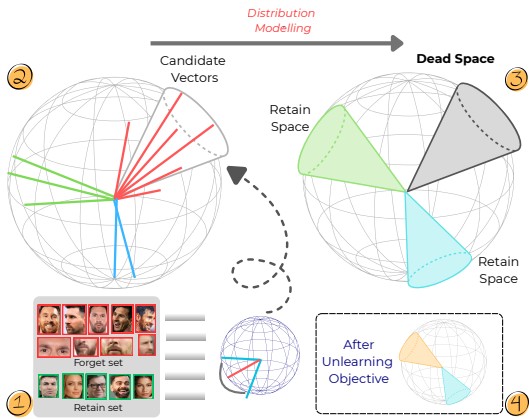

Figure 2: Structural erasure flow of FUSE illustrating 'dead space' formation using surrogate representation.

We model the embedding distribution of each identity $k \in Y$ as a spherical cone in the embedding space, described by two key parameters: a cone center that captures the most representative direction of the identity, and a cone spread that quantifies the variability of samples around this center.

**Cone Center (Modal Direction):** For an identity $k$, let $\mathcal{S}_k = \{i : y_i = k\}$ denote the set of samples belonging to it, with embeddings $\{s_i\}$. The cone center is simply the mean direction of all embeddings:

$$\mu_k = \frac{1}{|\mathcal{S}_k|} \sum_{i \in \mathcal{S}_k} s_i, \quad \mu_k \leftarrow \frac{\mu_k}{\|\mu_k\|_2}$$

This normalized vector $\mu_y$ represents the "axis" of the identity cone, pointing toward the most typical embedding location for class $k$.

**Cone Spread (Angular Deviation):** Not all embeddings perfectly align with the center. To measure how much they deviate, we compute the angular distance between each embedding $s_i$ and the cone axis $\mu_k$: $\theta_i = \arccos(\langle s_i, \mu_k \rangle)$. The overall spread is defined as the root-mean-square angular deviation:

$$\sigma_k = \sqrt{\frac{1}{|\mathcal{S}_k|} \sum_{i \in \mathcal{S}_k} \theta_i^2}$$

Here, $\sigma_k$ captures how tightly or loosely the embeddings of identity y cluster around their cone center. A smaller spread indicates a compact, well-defined identity cluster, whereas a larger spread corresponds to greater intra-class variability.

**Probabilistic Cone Model:** To capture the variability of embeddings within each identity cone, we model their angular distribution using a wrapped normal distribution on the hypersphere. Intuitively, this distribution treats deviations from the cone center as angular noise, while respecting the spherical geometry of the embedding space. For an embedding $s_i$ associated with identity $k : k \in Y$, the log-probability of belonging to the cone is given by:

$$\log p(s_i \mid k) \propto -\frac{1}{2\sigma_k^2} [\theta]^2, \quad \mathrm{LL}_{\text{proxy}}(s_i; k) = \exp\left(-\frac{\theta^2}{2\sigma_k^2}\right) \in (0, 1]$$

This probabilistic formulation provides a tractable density model over embeddings, where identities with small $\sigma_k$ form sharply concentrated cones, while those with larger spreads capture more intra-class variation. We approximate the likelihood of an embedding $s_i$ under class $k$ by a Gaussian-like function of its angular deviation from the class center. Formally, we use the unnormalized log-likelihood model $\log p(s_i \mid k)$. For practical evaluation, we define a bounded proxy likelihood score $LL_{\text{proxy}}$ which provides a tractable similarity measure even though it is not a fully normalized density on the hypersphere.

## 3.4 GEOMETRY-AWARE ERASURE

We proposed a region-based loss function for the purpose of trace removal from the feature space.

**Cone Repulsion Loss:** To explicitly erase the representation of a target identity $Y_u$, we introduce a cone repulsion loss that forces embeddings to move away from their original identity cone. The key idea is that samples belonging to the forgotten identity should no longer remain within the angular region characterized by their cone center and spread.

Let $\mathcal{S}_{Y_u}$ denote the set of samples from the target identity, and let $\tilde{s}_i$ be their modified embeddings under the unlearned model. For each sample, we measure its angular distance from the original cone axis $\mu_{Y_u}$. The repulsion loss is defined as:

$$\mathcal{L}_{\text{cr}} = \frac{1}{|\mathcal{S}_{Y_u}|} \sum_{i \in \mathcal{S}_{Y_u}} \max\left(0, \ \alpha\sigma_{Y_u} - \arccos(\langle \tilde{s}_i, \mu_{Y_u} \rangle)\right),$$

where $\sigma_{Y_u}$ is the spread of the original cone, and $\alpha > 0$ is a repulsion factor (e.g., $\alpha = 3$) that enforces a 3-sigma repulsion rule.

Intuitively, this loss penalizes embeddings that remain within the angular radius $\alpha\sigma_{Y_u}$ of the original cone center. By pushing samples outside this region, the model ensures that the samples lie away from the original cone distribution. Instead, these embeddings migrate into unrelated or noisy regions of the hypersphere, thereby destroying the semantic coherence of the target identity. Geometrically, the repulsion loss carves out a *dead space* around the original cone axis, forcing target identity embeddings to relocate beyond the angular boundary $\alpha\sigma_{Y_u}$, ensuring that the erased identity cannot be easily reconstructed.

**Cone Preservation Loss:** While the repulsion loss enforces forgetting for the target identity, it is equally important to preserve the structural integrity of all non-target identities. To achieve this, we introduce a cone preservation loss that maintains the original angular distribution of embeddings for $k \neq Y_u$. Formally, let $p_{\text{orig}}(s \mid k)$ and $p_{\text{FUSE}}(\tilde{s} \mid k)$ denote the original and modified angular distributions for identity k. We measure their divergence using the Kullback–Leibler (KL) divergence,

aggregated over all non-target classes:

$$\mathcal{L}_{\text{cp}} = \sum k \neq Y_u \frac{|\mathcal{S}_k|}{|\mathcal{D}_{tr}|} D_{\text{KL}}\big(p_{\text{orig}}(s \mid k) \,\|\, p_{\text{FUSE}}(\tilde{s} \mid k)\big)$$

where $|\mathcal{D}_{tr}|$ is the total dataset size and the weighting factor $|\mathcal{S}_k|/|\mathcal{D}_{tr}|$ ensures that identities with fewer samples are not underrepresented. The KL divergence can be approximated using the wrapped normal cone model, and substituting the wrapped normal form yields a tractable angular expression:

$$D_{\text{KL}}(p_{\text{orig}} \,\|\, p_{\text{FUSE}}) = \frac{1}{|\mathcal{S}_k|} \sum i \in \mathcal{S}_k \Big[ \log p_{\text{orig}}(s_i \mid k) - \log p_{\text{FUSE}}(\tilde{s}_i \mid k) \Big]$$

$$D_{\text{KL}}(p_{\text{orig}} \,\|\, p_{\text{FUSE}}) = \frac{1}{2|\mathcal{S}_k|} \sum i \in \mathcal{S}_k \left[ \frac{\big(\arccos(\langle \tilde{s}_i, \mu_k \rangle)\big)^2}{\sigma_k^2} - \frac{\big(\arccos(\langle s_i, \mu_k \rangle)\big)^2}{\sigma_k^2} \right]$$

This formulation penalizes deviations of modified embeddings from their original cone structure, effectively locking non-target identities to their characteristic angular distributions. As a result, while the target identity is dispersed and forgotten, the remaining identities remain stable and discriminative, preserving recognition performance for the retain set.

**Global Loss:** In addition to preserving individual identity cones, it is important to maintain the global geometric structure of the embedding space. Drastic modifications during unlearning may unintentionally distort inter-class relationships, leading to degraded discriminability among the retained identities. To mitigate this, we introduce a global distribution consistency loss that enforces stability of pairwise angular relationships across the entire dataset.

Formally, let $\{s_i\}$ denote the original embeddings and $\{\tilde{s}_i\}$ are the corresponding modified embeddings after unlearning. We define the Global loss as:

$$\mathcal{L}_{\text{g}} = \frac{1}{|\mathcal{D}_{tr} \neq Y_u|^2} \sum i, j \Big[ \langle s_i, s_j \rangle - \langle \tilde{s}_i, \tilde{s}_j \rangle \Big]^2,$$

where $\langle s_i, s_j \rangle$ denotes the cosine similarity between embeddings $s_i$ and $s_j$.

This objective penalizes discrepancies between the original and modified similarity structures, thereby ensuring that the relative angular positioning of embeddings remains intact. In effect, the global distribution consistency loss acts as a regularizer, while the repulsion and preservation losses operate locally at the level of specific identities. This term prevents large-scale distortions in the overall feature space topology, safeguarding discriminability among the retained set and stability of inter-class geometry. The complete unlearning objective combines all three loss components: $\mathcal{L}_{\text{total}} = \lambda_1 \mathcal{L}_{\text{cr}} + \lambda_2 \mathcal{L}_{\text{cp}} + \lambda_3 \mathcal{L}_{\text{g}}$ where $\lambda_1, \lambda_2, \lambda_3 > 0$ are weighting hyperparameters that control the relative importance of each objective.

## 4 RESULTS

We evaluate our method on two large-scale publicly available datasets: CASIA-WebFace and D-LORD and LFW. All implementation and dataset details are provided in Appendix E. In this section, we provide detailed experiments and analysis for our proposed unlearning method.

**Baselines:** We use existing unlearning methods trained on same protocol for fair comparison. We leverage Fine Tuning (Warnecke et al. (2021); Golatkar et al. (2020)), Gradient Ascent (GA) (Thudi et al. (2022)), $l_1$-sparse (Jia et al. (2023)), Random Labeling (RL) (Hayase et al. (2020)), Boundary Shrinking (BS), Boundary Expansion (BE) (Chen et al. (2023)), SalUn (Fan et al. (2024)) and SG-unlearn (Di et al. (2025)) for efficient comparison.

**Evaluation Criteria:** We evaluate unlearning effectiveness across three complementary dimensions. First, for membership analysis, we measure membership inference attack (MIA) accuracy Carlini et al. (2022), where effective unlearning reduces performance to near-random ( 0.5). Second, for feature-level evaluation, we compute verification accuracy (VA) using feature similarity: $V_r$ for retain classes, $V_f$ for the forget set, and $V_{uc}$ for unseen classes. The absolute difference $|V_f - V_{uc}|$ indicates whether forgotten identities are transformed into "unknown" identities. We also assess class drift for feature-level analysis (see Appendix D.2).

Table 1: Quantitative comparison of FUSE with existing unlearning techniques. VA represents Verification Accuracy, and SR represents Semantic Residual. All experiments are performed on the ResNet-50 backbone initially trained with ArcFace loss.

| Model | CASIA-Webface (Yi et al. (2014)) | | | | | | | D-LORD (Manchanda et al. (2023)) | | | | | | |
|---|---|---|---|---|---|---|---|---|---|---|---|---|---|---|
| | MIAa | VA | | | | SR | | MIAa | VA | | | | SR | |
| | | $V_r$ | $V_f$ | $V_{uc}$ | $V_f - V_{uc}$ | $G_r$ | $G_f$ | | $V_r$ | $V_f$ | $V_{uc}$ | $V_f - V_{uc}$ | $G_r$ | $G_f$ |
| OG | 0.903 | 0.926 | 0.836 | 0.867 | 0.031 | 1.000 | 0.000 | 0.784 | 0.847 | 0.795 | 0.814 | 0.019 | 1.000 | 0.000 |
| FT | 0.582 | 0.863 | 0.802 | 0.752 | 0.050 | 0.420 | 0.385 | 0.539 | 0.782 | 0.784 | 0.797 | 0.013 | 0.385 | 0.372 |
| GA (Thudi et al. (2022)) | 0.621 | 0.834 | 0.518 | 0.774 | 0.256 | 0.488 | 0.643 | 0.692 | 0.749 | 0.519 | 0.638 | 0.119 | 0.362 | 0.374 |
| L1-sparse (Jia et al. (2023)) | 0.572 | 0.795 | 0.647 | 0.721 | 0.074 | 0.540 | 0.648 | 0.493 | 0.751 | 0.558 | 0.630 | 0.072 | 0.442 | 0.401 |
| RL (Hayase et al. (2020)) | 0.539 | 0.815 | 0.692 | 0.737 | 0.045 | 0.559 | 0.539 | 0.512 | 0.792 | 0.539 | 0.619 | 0.080 | 0.410 | 0.485 |
| BS (Chen et al. (2023)) | 0.562 | 0.827 | 0.684 | 0.793 | 0.109 | 0.532 | 0.572 | 0.568 | 0.763 | 0.603 | 0.623 | 0.020 | 0.386 | 0.403 |
| BE (Chen et al. (2023)) | 0.429 | 0.813 | 0.659 | 0.693 | 0.034 | 0.561 | 0.537 | 0.482 | 0.683 | 0.581 | 0.554 | 0.027 | 0.404 | 0.395 |
| SalUn (Fan et al. (2024)) | 0.485 | 0.858 | 0.599 | 0.649 | 0.050 | 0.498 | 0.487 | 0.441 | 0.774 | 0.597 | 0.672 | 0.075 | 0.412 | 0.382 |
| SG (Di et al. (2025)) | 0.503 | 0.849 | 0.680 | 0.728 | 0.048 | 0.525 | 0.461 | 0.381 | 0.753 | 0.606 | 0.642 | 0.036 | 0.481 | 0.342 |
| FUSE (ours) | 0.519 | 0.851 | 0.739 | 0.751 | 0.012 | 0.632 | 0.391 | 0.461 | 0.780 | 0.635 | 0.653 | 0.018 | 0.538 | 0.328 |

Table 2: Quantitative comparison of FUSE on WebFace4m and MS1Mv3 datasets.

| Model | WebFace4M (Zhu et al. (2022)) | | | | | | | MS1Mv3 (Liu et al. (2015)) | | | | | | |
|---|---|---|---|---|---|---|---|---|---|---|---|---|---|---|
| | $MIA_a$ | VA | | | | SR | | $MIA_a$ | VA | | | | SR | |
| | | $V_r$ | $V_f$ | $V_{uc}$ | $V_f - V_{uc}$ | $G_r$ | $G_f$ | | $V_r$ | $V_f$ | $V_{uc}$ | $V_f - V_{uc}$ | $G_r$ | $G_f$ |
| OG | 0.88 | 0.89 | 0.81 | 0.82 | 0.01 | - | - | 0.89 | 0.91 | 0.92 | 0.88 | 0.04 | - | - |
| FT | 0.53 | 0.79 | 0.61 | 0.76 | 0.15 | 0.21 | 0.57 | 0.46 | 0.72 | 0.80 | 0.78 | 0.02 | 0.42 | 0.37 |
| GA (Thudi et al. (2022)) | 0.64 | 0.75 | 0.65 | 0.78 | 0.13 | 0.25 | 0.44 | 0.51 | 0.77 | 0.64 | 0.79 | 0.15 | 0.45 | 0.51 |
| L1-sparse (Jia et al. (2023)) | 0.60 | 0.78 | 0.69 | 0.74 | 0.05 | 0.31 | 0.48 | 0.55 | 0.78 | 0.72 | 0.81 | 0.09 | 0.49 | 0.47 |
| RL (Hayase et al. (2020)) | 0.70 | 0.77 | 0.67 | 0.69 | 0.02 | 0.35 | 0.51 | 0.49 | 0.78 | 0.61 | 0.72 | 0.11 | 0.48 | 0.48 |
| BS (Chen et al. (2023)) | 0.61 | 0.81 | 0.70 | 0.76 | 0.06 | 0.28 | 0.55 | 0.67 | 0.81 | 0.76 | 0.76 | 0.00 | 0.39 | 0.40 |
| BE (Chen et al. (2023)) | 0.44 | 0.80 | 0.71 | 0.69 | 0.02 | 0.38 | 0.49 | 0.52 | 0.75 | 0.78 | 0.75 | 0.03 | 0.50 | 0.41 |
| SalUn (Fan et al. (2024)) | 0.48 | 0.77 | 0.68 | 0.71 | 0.03 | 0.33 | 0.51 | 0.48 | 0.78 | 0.81 | 0.72 | 0.09 | 0.51 | 0.36 |
| SG (Di et al. (2025)) | 0.52 | 0.80 | 0.67 | 0.73 | 0.06 | 0.38 | 0.43 | 0.41 | 0.76 | 0.73 | 0.78 | 0.05 | 0.47 | 0.33 |
| FUSE (ours) | 0.53 | 0.83 | 0.75 | 0.79 | 0.04 | 0.43 | 0.42 | 0.47 | 0.85 | 0.80 | 0.80 | 0.00 | 0.58 | 0.29 |

*Semantic Evaluation:* We further propose **Semantic Residual (SR)** auditing (see Appendix D.1), a generation-free, score-matching based evaluation of privacy leakage. Unlike generative auditing, which relies on noisy reconstructions and uncertain classification, SR directly quantifies residual identity traces in the embedding space without image synthesis. For a given condition c, we define:

$$\text{SR}(c) = \min(1, \frac{\tilde{L}\text{score}}{L\text{score}}), \quad L_{\text{score}} = \mathbb{E}\big[\|\epsilon_\theta(x_t, t, c) - \epsilon_{\text{true}}\|^2\big],$$

where $L_{\text{score}}$ and $\tilde{L}_{\text{score}}$ are computed under the original and unlearned distributions, respectively. We report SR for retain ($G_r$) and forget ($G_f$) sets, where high $G_f$ indicates semantic leakage (unlearning failure), and low $G_f$ reflects effective forgetting while maintaining discriminability.

**Class Drift:** For samples belonging to the forget class, our objective is to quantify how strongly they still align with their original class distribution after unlearning. Since the classifier proxy (e.g., class weight vector) is no longer available, we instead measure the average distributional shift of a class relative to the original cone distribution. The mathematical formulation of class drift in presented in Appendix D.2. The class drift values lie between 0 and 1. The value decreases monotonically as the angle between the class samples and cone $\theta$ increases, approaching 0 as z drifts far from the class cone. A low class drift suggests that the embedding has drifted away from the original class cone, implying more effective unlearning. We can see from the Figure 3, our method achieves lowest value, showing highest drift of class distribution.

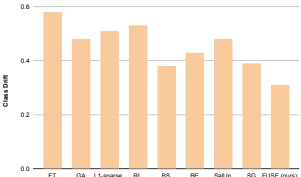

Figure 3: Comparison of the average distributional shift of forget set class w.r.t. to the original distribution.

**Quantitative Results:** Table 2 highlights trade-offs among existing unlearning methods. The baseline (OG) shows high MIA and verification on the forget set, indicating no forgetting. Fine-tuning (FT) reduces MIA but at the cost of retain utility ($V_r$). Gradient-based methods (GA, L1-sparse) achieve partial forgetting but leave a large $V_f - V_{uc}$ gap. SalUn and SG balance forgetting and retention better but still leak semantic information (higher $G_f$). In contrast, FUSE attains near-random MIA and preserves verification on seen ($V_r$) and unseen ($V_{uc}$) classes, minimizes $|V_f - V_{uc}|$. For semantic residual as well, FUSE achieves the highest $G_r$ (retain set) and the lowest $G_f$ (forget set).

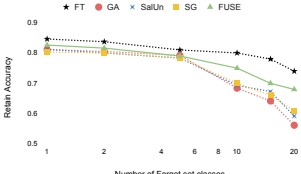

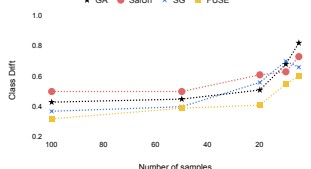

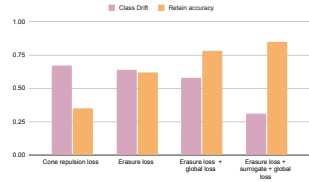

(a) Number of forget set classes     (b) Sample size of forget class     (c) Effect of various components

Figure 4: Ablation study of FUSE. (a) Retain accuracy vs. number of forget classes, showing FUSE maintains higher performance under increasing forget sets. (b) Class drift vs. forget class size, highlighting robustness with smaller classes. (c) Effect of individual components for unlearning.

**Ablation Study:** Figure 4 presents an ablation study on the effect of different factors in FUSE. In Figure 4a, as the number of forget classes increases, all methods show reduced retain accuracy, but FUSE consistently preserves higher performance, demonstrating more selective forgetting.

Table 3: Performance Comparison of targeted loss with original cone repulsion loss.

| | Verification Accuracy | | semantic residuals | |
|---|---|---|---|---|
| | $V_r$ | $V_f$ | $G_r$ | $G_f$ |
| Untargeted loss (Paper) | 0.85 | 0.74 | 0.63 | 0.39 |
| Targeted loss (New) | 0.84 | 0.75 | 0.64 | 0.41 |

Figure 4b shows that smaller forget class sizes cause larger drift due to unstable class modeling, with FUSE achieves the lowest drift overall, indicating robustness of its cone–based design.

Figure 4c examines individual components. Cone repulsion alone provides partial forgetting but hurts retention, while adding preservation loss improves balance. The best trade-off is achieved when global and surrogate losses are included, confirming each component contributes meaningfully, with the FUSE framework achieving the stable and effective unlearning. Further, we also show comparison of existing $L_c r$ with targeted cone repusion loss. The results are shown in Table 3. More details of this targeted loss are present in Appendix B. Hyperparameter ablation is shown in Table 4. Refer Appendix F for more results.

Table 4: Ablation study on hyperparameters.

| Hyperparameter Configuration | semantic residuals | |
|---|---|---|
| | $G_r$ | $G_f$ |
| $\lambda_1 = 1, \lambda_2 = 1, \lambda_3 = 0.1$ | 0.58 | 0.29 |
| $\lambda_1 = 0.5, \lambda_2 = 1, \lambda_3 = 0.2$ | 0.56 | 0.48 |
| $\lambda_1 = 1, \lambda_2 = 0.5, \lambda_3 = 0.1$ | 0.46 | 0.33 |
| $\lambda_1 = 0.8, \lambda_2 = 1, \lambda_3 = 0.1$ | 0.56 | 0.44 |
| $\lambda_1 = 1, \lambda_2 = 0.8, \lambda_3 = 0.1$ | 0.50 | 0.32 |
| $\lambda_1 = 1, \lambda_2 = 1, \lambda_3 = 0.2$ | 0.54 | 0.29 |

**Robustness of FUSE:** We evaluate the effectiveness of FUSE under three attack settings: adversarial robustness, inversion attacks, and membership inference. For adversarial reconstruction, the attacker attempts to extract any remaining information about the forgotten class. Our semantic residual metric serves as a proxy for this attack by quantifying how much class-specific structure can still be recovered from the feature space, without requiring explicit reconstruction. Using three generative models, we compute semantic residuals for both retain and forget sets.

Table 5: Comparative evaluation for various generative models.

| Model | Semantic Residual | | | | | |
|---|---|---|---|---|---|---|
| | IDiff-Face | | Arc2Face | | Guided Diffusion | |
| | $G_r$ | $G_f$ | $G_r$ | $G_f$ | $G_r$ | $G_f$ |
| FT | 0.385 | 0.372 | 0.315 | 0.482 | 0.327 | 0.385 |
| GA | 0.362 | 0.374 | 0.450 | 0.505 | 0.363 | 0.338 |
| L1-sparse | 0.442 | 0.401 | 0.385 | 0.474 | 0.432 | 0.431 |
| RL | 0.410 | 0.485 | 0.483 | 0.414 | 0.385 | 0.453 |
| BS | 0.386 | 0.403 | 0.373 | 0.305 | 0.382 | 0.385 |
| BE | 0.404 | 0.395 | 0.528 | 0.494 | 0.428 | 0.385 |
| SalUn | 0.412 | 0.382 | 0.489 | 0.392 | 0.404 | 0.337 |
| SG | 0.481 | 0.342 | 0.510 | 0.292 | 0.452 | 0.402 |
| FUSE (ours) | 0.538 | 0.328 | 0.639 | 0.274 | 0.496 | 0.331 |

Across all settings, retain-set residuals remain stable—indicating preserved utility—while FUSE consistently produces the lowest residuals for the forget set. This confirms that FUSE effectively removes class-specific information from the unlearned identities.

Inversion Attack: Given an unlearned model, inversion attack produce images that reveal the identity/appearance of the forgotten class. In this attack, we applied optimization-based attack, where we assume white-box setting, and we have access to the mean embedding of forgotten class. If the attacker knows the target embedding (e.g., the forgotten class centroid), they can optimize the image so that: $f(x_{\text{generated}}) \approx \text{target\_embedding}$. This is essentially "searching" for an image whose features match the forgotten identity. We initialize with the random noise, and we apply the loss: $\mathcal{L}_{\text{feat}} = \|f(x) - \text{target\_embedding}\|^2$. This forces the reconstruction to match the identity features of the forgotten class. After getting such pool of embeddings, we compute Semantic residual $G_f$ for the forgotten class to see if it reaches any close to its original class distribution - if the attack is successful. The results are shown in Appendix F.

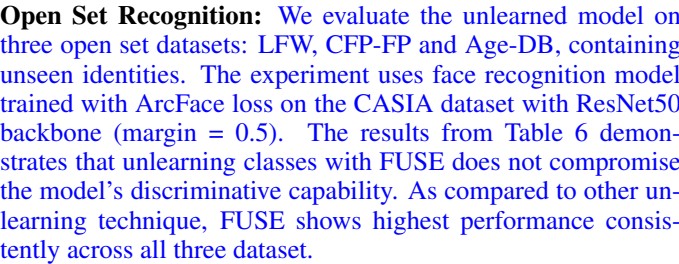

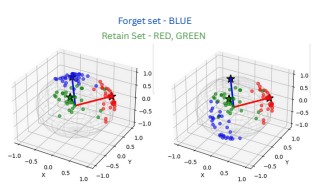

Figure 6: Showcasing reconstructed facial samples generated by IDiff-Face reconstruction model. Illustrating that identities forgotten through FUSE cannot be reconstructed, whereas other unlearning methods still allow facial reconstruction. FUSE also preserves the identities in the retain set, as seen in the samples in green block.

Figure 7: Feature Representation before and after unlearning.

**Feature representation:** We visualize embedding distributions of unseen samples before and after unlearning to assess FUSE's impact on class representation (Figure 7). Before unlearning, unseen samples form compact clusters around their class centers, reflecting strong discriminability. After FUSE, retain set samples remain tightly clustered around their means, showing high retention, while forget set samples also form coherent clusters but are no longer aligned with their original centers. FUSE thus achieves unlearning at the representation level—disrupting identity-specific alignment while maintaining discriminative capacity.

**Hard Set Retention:** Owing to the fine-grained nature of face recognition, unlearning can disproportionately affect closely related classes. To study this, we construct a hard set by retrieving the top-$k$ nearest classes from the training set using cosine similarity, with $k = 10$ in our experiments. For these classes, we evaluate both the average retain accuracy and the class drift (higher values are better). Results (Figure 5) show that retaining performance on the hard set is significantly more challenging compared to the overall dataset. However, due to its geometry-aware preservation of retain class distributions, FUSE achieves consistently higher retention on the hard set than competing approaches, demonstrating superior robustness in fine-grained scenarios.

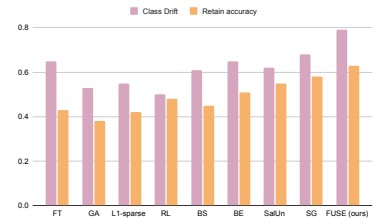

Figure 5: Performance evaluation (Class drift and retention accuracy - higher is better) on top-10 hard set.

**Open Set Recognition:** We evaluate the unlearned model on three open set datasets: LFW, CFP-FP and Age-DB, containing unseen identities. The experiment uses face recognition model trained with ArcFace loss on the CASIA dataset with ResNet50 backbone (margin = 0.5). The results from Table 6 demonstrates that unlearning classes with FUSE does not compromise the model's discriminative capability. As compared to other unlearning technique, FUSE shows highest performance consistently across all three dataset.

Table 6: Open set recognition performance on three datasets.

|  | LFW | CFP-FP | Age-DB |
|---|---|---|---|
| Original | 99.53 | 95.54 | 95.15 |
| GA | 83.25 | 91.36 | 93.01 |
| L1-sparse | 82.44 | 90.53 | 92.74 |
| RL | 91.12 | 91.46 | 94.73 |
| BS | 85.47 | 92.04 | 94.02 |
| BE | 87.97 | 92.58 | 93.59 |
| SalUn | 93.59 | 91.99 | 94.58 |
| SG | 98.17 | 92.16 | 95.12 |
| FUSE (ours) | 99.31 | 92.97 | 95.12 |

## 5 CONCLUSION

In this paper, we explore a fundamental aspect of machine unlearning for discriminative models such as face recognition systems. We formalize unlearning in the context of fine-grained classification, where an ideal unlearning method must preserve high discriminability for unseen classes and unseen samples of forgotten classes, thereby maintaining generalization, while simultaneously erasing semantic traces of the target identity. Unlike prior approaches that primarily rely on verification accuracy, which inherently reflects retention rather than forgetting, we propose to evaluate unlearning through the persistence of semantic information in the feature space. We introduce FUSE, a method that erases the distribution of a target class by dispersing its feature cone, effectively removing residual semantics while safeguarding the representational integrity of identities. Our experimental results demonstrate that FUSE preserves generalization capability while achieving semantic forgetting, establishing a novel and necessary direction for unlearning in discriminative models.

ETHICAL CONSIDERATION

This work does not involve potential malicious or unintended uses, fairness considerations, privacy considerations, security considerations and crowd sourcing. All experiments on facial images are performed on publicly available datasets.

REPRODUCIBILITY STATEMENT

We provide details to reproduce our results with all implementation and training details mentioned in the Appendix E. We will release the code upon acceptance.

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

## APPENDIX

## A   REPULSION FACTOR

To formalize the repulsion margin in our cone-based unlearning framework, we draw upon the well-established 3-sigma rule of Gaussian distributions. For a random variable $X \sim \mathcal{N}(\mu, \sigma^2)$, approximately 68% of the probability mass lies within one standard deviation ($\mu \pm \sigma$), 95% within two ($\mu \pm 2\sigma$), and 99.7% within three ($\mu \pm 3\sigma$). Consequently, the probability of observing a sample outside the $\mu \pm 3\sigma$ interval is less than 0.3%. This property provides a natural statistical threshold for distinguishing inliers from outliers in a Gaussian-like distribution.

In our setting, the angular distribution of embeddings around the class center $\mu_y$ can be modeled using a wrapped normal distribution. Here, $\sigma_y$ represents the angular spread of embeddings within the cone. By enforcing a repulsion margin of at least $3\sigma_y$, we effectively push the forgotten class embeddings into regions of the hypersphere that lie outside the statistically plausible support of their original distribution. This guarantees, with high probability, that embeddings no longer remain consistent with the identity they once represented. At the same time, embeddings of retain classes are preserved through cone-preservation and global consistency losses, ensuring that the discriminative geometry of the feature space remains intact.

This statistical perspective highlights why the 3-sigma margin is not an arbitrary design choice but a principled boundary for unlearning. By displacing forgotten embeddings beyond the 99.7% confidence region of their original cone, we ensure that the model no longer encodes those identities in a semantically valid form, while retaining the broader discriminative power of the feature extractor. Thus, the 3-sigma rule provides both a theoretical guarantee of forgetting and a geometric interpretation of how far embeddings must be displaced to erase identity traces effectively.

## B  TARGETED REPULSION LOSS

FUSE is intentionally designed to dissolve the identity geometry rather than relocate forgotten samples to a fixed region, since forcing them into a predetermined cluster may inadvertently create a new identity-like structure. Our original goal is to erase identity-specific information by removing the semantic coherence of the original cone while preserving the geometry of all retained classes.

To further address the reviewer's concern, we conducted an additional experiment where we augment FUSE with a uniform-diffusion loss. This term encourages the forgotten embeddings to spread toward an isotropic, structureless distribution on the hypersphere, that aligns with the motivation of FUSE:

$$L_{\text{targeted}} = \frac{1}{|S_{Y_u}|} \sum_i \theta_i, \quad \theta_i = \arccos\left(\langle \hat{s}_i, u_i \rangle\right)$$

where each $u_i$ is a randomly sampled unit vector drawn uniformly from the hypersphere. This controlled diffusion prevents forgotten embeddings from drifting arbitrarily, ensuring that they migrate toward a stable, non-informative, and identity-free region of the feature space.

Conceptually, this aligns with the purpose of targeted unlearning: instead of forming a new cluster, forgotten samples are dissolved into a uniform noise landscape, their identity structure becomes irrecoverable, and they no longer interfere with retained identities.

Empirically, we observe that adding this loss produces a structured form of forgetting while maintaining the stability of the retained classes. We will include this additional variant as an ablation to demonstrate that FUSE can incorporate explicit destination control when desired, although our main formulation intentionally avoids imposing a new identity structure on forgotten samples.

## C  HYPERSPHERE-CONSTRAINED OPTIMIZATION FOR UNLEARNING

In face recognition, embeddings are typically constrained to lie on the unit hypersphere. This constraint ensures compatibility with popular margin-based objectives such as ArcFace or CosFace, which assume normalized feature vectors for stable angular comparisons. When performing unlearning, it is therefore essential to maintain this geometric property: embeddings of both retained and forgotten identities must remain normalized while undergoing repulsion, preservation, and consistency updates. To achieve this, we formulate unlearning as a constrained optimization problem on the hypersphere and solve it using a projected gradient descent scheme.

1. **Hypersphere Constraint:**  Each modified embedding is explicitly normalized to unit length:

$$\|\tilde{z}_i\|_2 = 1 \quad \forall i, \qquad \tilde{z}_i \leftarrow \frac{\tilde{z}_i}{\|\tilde{z}_i\|_2}.$$

2. **Optimization Algorithm:**  The total unlearning objective, $\mathcal{L}$, combines repulsion, preservation, and consistency losses. We optimize it while ensuring that embeddings remain on the hypersphere:

   - Gradient Computation: $g_i = \nabla_{\tilde{z}_i} \mathcal{L}_{\text{total}}$
   - Tangent Space Projection: project the gradient onto the tangent plane of the hypersphere to prevent updates that move embeddings off the sphere:

$$g_i^{\text{tangent}} = g_i - (\tilde{z}_i^\top g_i) \tilde{z}_i$$

   - Gradient Update: $\tilde{z}_i^{(t+1)} = \tilde{z}_i^{(t)} - \eta\, g_i^{\text{tangent}}$
   - Sphere Projection: renormalize the embedding after each step to strictly enforce the hypersphere constraint:

$$\tilde{z}_i^{(t+1)} \leftarrow \frac{\tilde{z}_i^{(t+1)}}{\|\tilde{z}_i^{(t+1)}\|_2}$$

## C.1 THEORETICAL ANALYSIS

We now provide a theoretical analysis of our hypersphere-constrained unlearning framework, focusing on convergence guarantees, angular preservation properties, and forgetting effectiveness. This analysis demonstrates that the proposed objectives are not only geometrically meaningful but also enjoy favorable optimization properties under mild assumptions.

***Proposition 1 (Local Convergence).*** Under standard regularity conditions (Lipschitz-continuous gradients and a bounded feasible set), the projected gradient descent algorithm converges to a local minimum of the constrained optimization problem.

*Proof Sketch.* The feasible set of embeddings lies on the unit hypersphere, which is a smooth Riemannian manifold. Since the loss functions (repulsion, preservation, consistency) are differentiable, the projected gradient method coincides with Riemannian gradient descent. Existing results in optimization on Riemannian manifolds guarantee convergence to stationary points under standard assumptions, similar to the Euclidean setting.

***Proposition 2 (Inter-Class Angular Preservation).*** For any pair of non-target identities $y_i, y_j \neq y_{\text{target}}$, the angular distortion between their cone centers after unlearning is bounded as

$$\left| \langle \mu_{y_i}, \mu_{y_j} \rangle - \langle \tilde{\mu}_{y_i}, \tilde{\mu}_{y_j} \rangle \right| \leq \epsilon,$$

where $\epsilon$ depends on the magnitude of embedding modifications and the weight of the consistency loss $\lambda_3$.

*Proof Sketch.* The global consistency loss penalizes deviations in pairwise cosine similarities between embeddings. As $\lambda_3$ increases, the optimizer is forced to preserve inter-class angular relationships more faithfully. Thus, in the limit of large $\lambda_3$, angular distortions among non-target classes vanish.

***Proposition 3 (Target Identity Separation).*** At convergence, the embeddings of the target identity satisfy the separation condition

$$\min_{i \in \mathcal{S} y_{\text{target}}} \arccos\left( \langle \tilde{z}_i, \mu_{y_{\text{target}}} \rangle \right) \geq \alpha \sigma_{y_{\text{target}}} - \delta,$$

where $\delta \to 0$ as the repulsion weight $\lambda_1 \to \infty$.

This bound guarantees that all target identity embeddings are pushed outside their original cone by at least the specified angular margin, ensuring that the forgotten identity cannot retain a compact or recoverable cluster.

**Computational Complexity**

The computational bottleneck arises from the global consistency loss, which requires pairwise similarity computations across the dataset, scaling as $\mathcal{O}(|\mathcal{D}|^2)$. For large datasets, this cost can be mitigated by subsampling pairs, adopting mini-batch approximations, or using structured regularizers that approximate global similarity preservation.

# D EVALUATION METHODS

## D.1 SEMANTIC RESIDUAL

To compute how much class-related information is held by the new distribution (from the unlearned model) for the forget class, we leverage a diffusion-based generative model. This model is pretrained on the original model (before unlearning), taking the original class distribution as conditional guidance. Since the distribution is changed for the unlearned model for the forgotten class, the conditions will also be changed. Thus, for this changed distribution, how much the diffusion model knows what to generate is computed using semantic residual.

For any denoising diffusion model, we compute the mean squared error (MSE) between the predicted noise and the ground-truth noise under both the original and modified conditioning distributions.

From the original distribution, the score matching error is: From original distribution:

$$L_{\text{score}} = \mathbb{E}c \sim p_{\text{orig}}\big[\|\epsilon_\theta(x_t, t, c) - \epsilon_{\text{true}}\|^2\big]$$

From the modified (unlearned) distribution, it is defined as:

$$\tilde{L}_{\text{score}} = \mathbb{E}c \sim p_{\text{new}}\big[\|\epsilon_\theta(x_t, t, c) - \epsilon_{\text{true}}\|^2\big]$$

Finally, we define the Semantic Residual (SR) as the normalized reduction in score-matching error:

$$SR = \min(1, \frac{\tilde{L}_{\text{score}}}{L_{\text{score}}})$$

This ratio measures how much semantic information from the original distribution persists after unlearning.

The resulting range for SR is $SR \in [0, 1]$. The normalized SR metric represents the proportion of original performance retained after the unlearning process.

- if $SR = 0$ This is the ideal outcome, signifying perfect forgetting. It means the new model's performance on the class is zero, so no semantic information remains.

- $SR = 1$: This signifies no forgetting. The new model's performance is equal to the original, indicating that the unlearning process was ineffective.

- This indicates partial retention. The value of SR shows what percentage of the original performance on the class still exists in the unlearned model. For example, SR =0.5 means 50% of the original predictive power for that class remains.

A higher SR for the forget set ($G_f$) indicates stronger semantic leakage, while lower values demonstrate effective removal of identity traces. Conversely, stable values of SR on the retain set ($G_r$) confirm preservation of discriminative capacity. We take a denoising UNET, for any time step $t$, we add some noise as forward noising. Take a batch of conditions $c \sim p_{\text{new}}$ and $c \sim p_{\text{orig}}$ (e.g., embeddings of the unlearned model and original model for a class). For each condition, run the diffusion model at a timestep t and get the predicted noise $\epsilon_\theta(x_t, t, c)$. Then we compute the ratio of the MSE of the predicted and ground-truth noise $\epsilon_{\text{true}}$ added during the forward process.

$L_{\text{score}}$ measures how wrong the model is in predicting the noise trajectory under a given condition. But in diffusion, predicting the noise correctly = reconstructing the data distribution correctly. Thus, SR acts as a proxy for how much meaningful structure remains in the embedding from the unlearned distribution with respect to the original distribution. Mathematically, predicting noise is the same as estimating the gradient of the log-probability (the "score" of the data distribution):

$$\epsilon_\theta(x_t, t, c) \approx -\sqrt{1 - \bar{\alpha}_t}\,\nabla_x \log p(x_t|c).$$

So the model isn't just denoising—it's learning the geometry of the conditional data distribution.

- If SR is close to 1, embedding still drives accurate generation, with high semantic leakage (identity not forgotten).

- If SR is close to 0, the model cannot use the embedding to generate: successful unlearning.

## D.2 CLASS DRIFT

We aim to compute the average class distribution shift for the forget class. Formally, let $\mu_A$ denote the modal direction and $\sigma_A$ the angular spread of class A. For an embedding $z$, the angular distance from the class center is $\theta = \arccos(\langle z, \mu_A \rangle)$. The cone-based log-likelihood of z under class A is given by: $\log p(z \mid A) \propto -\frac{\theta^2}{2\sigma_A^2}$. Equivalently, we define the normalized likelihood score as:

$$\text{LL}(z; A) = \exp\Big(-\frac{\theta^2}{2\sigma_A^2}\Big).$$

A high $\text{LL}(z; A)$ indicates that the embedding remains highly consistent with the original distribution of class A. A low $\text{LL}(z; A)$ suggests that the embedding has drifted away from the original class cone, implying more effective unlearning.

We compute the aggregate of mean likelihood over the class samples $Y$:

$$\overline{\text{LL}}(A) \;=\; \frac{1}{Y} \sum_{z \in Y} \text{LL}(z; A).$$

Values near 1: embeddings remain tightly aligned with the original cone (strong trace). Values near 0: embeddings have large angular drift (weak or no trace).

## E    IMPLEMENTATION DETAILS

For all experiments, we use the ResNet-50 backbone, trained on the loss arcface and cosface. For training a face recognition model, two datasets are used: CASIA-WebFace and D-LORD. For CASIA-WebFace, 10,000 classes are used for training, and 575 are kept for unseen class testing. For D-LORD, 1600 are used for training, while 600 are used for unseen testing. The FR model is trained using a 128 batch size, a learning rate of $1e-4$, and using SGD as optimizer.

For training FUSE, training is performed for 20 epochs, and the forget set size varies for different experiments, as shown in the ablation study. For surrogate formation, the face image is divided into two parts, the upper face and the lower face. Repulsion Factor ($\alpha$). We recommend values in the range $\alpha \in [2, 4]$ to enforce meaningful separation while avoiding over-displacement that could distort nearby cones. Baselines are also trained with the same parameters for fair comparison.

Loss Weights. A balanced configuration is $\lambda_1 = 1.0$ (repulsion as primary driver), $\lambda_2 = 1.0$ (equal importance to preservation), and $\lambda_3 = 0.1$ (light regularization for global consistency).

For semantic residual (SR) evaluation, we use IDiff-Face (guided diffusion), taking class embeddings as conditions. We use test unseen samples from classes, take a random $t$ for denoising and compute error. The training of this diffusion is done using an initial ArcFace model without any unlearning, the training parameters are the same as in the main paper.

## F    RESULTS

### F.1    INVERSION ATTACK RESULTS

The results show that all baseline methods experience a substantial increase in semantic residual after the inversion attack, suggesting that some degree of forgotten-class information can still be recovered. Methods such as GA, L1-sparse, and SalUn show particularly large increases, indicating higher vulnerability. In contrast, FUSE exhibits the lowest semantic residual both before and after the attack, with only a modest increase from 0.391 to 0.423. This demonstrates that FUSE leaves behind minimal recoverable identity information and is significantly more robust to inversion attacks compared to other unlearning methods. The behavior confirms FUSE strong resistance against feature-space reconstruction attempts targeting the forgotten class.

Table 7: Forget set semantic residual before and after attack.

| Model | Semantic Residual | |
|---|---|---|
| | Before Attack | After Attack |
| FT | 0.385 | 0.524 |
| GA | 0.643 | 0.672 |
| L1-sparse | 0.648 | 0.732 |
| RL | 0.539 | 0.592 |
| BS | 0.572 | 0.619 |
| BE | 0.537 | 0.543 |
| SalUn | 0.487 | 0.582 |
| SG | 0.461 | 0.482 |
| FUSE (ours) | 0.391 | 0.423 |

### F.2    ABLATION ON VARIOUS BACKBONES

As shown in Figure 8 Across all three backbone architectures—ResNet-50 (a), ResNet-101 (b), and ViT (c)—a consistent trend emerges in terms of semantic residuals for both retained and forgotten classes. For the retained set ($G_r$), FUSE achieves the highest or near-highest values among all baselines, indicating that identity information for non-forgotten classes is well preserved. In contrast, for

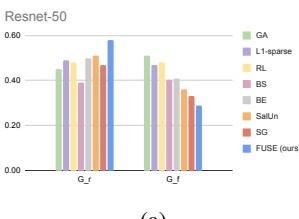 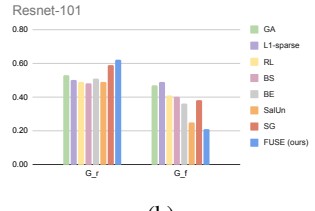 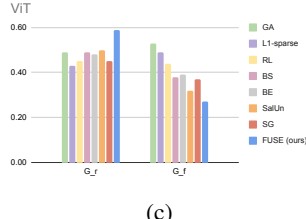

       (a)                             (b)                            (c)

Figure 8: Comparison of unlearning performance across three backbone architectures: ResNet-50 (a), ResNet-101 (b), and ViT (c). Each bar plot reports the geometric retention score for the retained class ($G_r$) and the forgotten class ($G_f$) under eight unlearning baselines, including GA, RL, BE, SalUn, SG, and our proposed FUSE.

the forgotten set ($G_f$), FUSE produces the lowest residuals across all architectures, demonstrating that it removes class-specific information more effectively than competing methods. Other baselines show varying degrees of unlearning, but none simultaneously achieve high $G_r$ and low $G_f$ as reliably as FUSE. This consistent separation highlights FUSE's strong ability to maintain utility while maximally erasing information from the forgotten identity classes.

### F.3 ABLATION ON HYPERPARAMETER

To analyze the model's behavior and understand the contribution of each loss component in FUSE, we conduct experiments by varying the hyperparameters, which weight the different loss terms. Using the MS1MV3 dataset, we evaluate the model across multiple configurations. In the paper, we adopt the setting $\lambda_1 = 1$, $\lambda_2 = 1$, $\lambda_3 = 0.1$, and the experimental results support this choice.

As shown in Table 4, when $\lambda_1$ is reduced, the repulsion applied to forgotten identities weakens. As a result, the retain-set geometry remains mostly unaffected, but forgotten identities remain close to their original cone, yielding a high $G_f$. Conversely, when $\lambda_2$ is reduced, the retain set drifts away from its original geometric structure, indicating lower perseverance. These observations show that an appropriate balance between the loss weights is necessary to achieve effective forgetting while preserving the retain set structure.

## G DISCUSSIONS

### G.1 COMPUTATIONAL EFFICIENCY

Our proposed FUSE training is computationally efficient, leveraging a small CNN-based feature extractor to obtain class-level representations. Our loss objectives consist of simple repulsion and preservance terms. To reduce the computational cost of the global loss, we compute it on a randomly selected subset of sample pairs in each training epoch. The surrogate data representation similarly employs a lightweight 3-layer MLP to generate face-related embeddings. For evaluating Semantic Residual, we avoid full denoising, which is costly, and instead compute the reconstruction error at a single random time step per sample, significantly reducing computation while retaining meaningful evaluation.

### G.2 LIMITATIONS

Our work with FUSE establishes a first step toward principled representation unlearning in face recognition, but several directions remain open for future research. The cone-based parameterization provides a tractable way to model identity distributions, yet future approaches could extend this to richer distributions that disentangle pose, age, and illumination, enabling more precise forgetting while preserving retention. Similarly, while FUSE leverages fixed hyperparameters for repulsion, preservation, and consistency, adaptive strategies for tuning these weights dynamically may improve stability across datasets and varying unlearning scales.

Beyond the algorithm itself, our evaluation framework, particularly Semantic Residual (SR), offers a generation-free measure of semantic leakage, but it can be expanded with stronger adversarial tests such as inversion and membership inference to provide more rigorous privacy guarantees. Extending the principles of distribution erasure and semantic auditing beyond faces to other fine-grained tasks like text, voice, or multimodal models may uncover broader foundations for representation-level unlearning.

