# OpenReview forum: "Unlearning Paradox: Auditing Residual Identity Traces in Face Recognition"
_ICLR.cc/2026/Conference — ICLR 2026 Conference Desk Rejected Submission_

### Official Review · Reviewer_tEMv · 2025-10-16

**Soundness:** 4
**Presentation:** 4
**Contribution:** 4
**Rating:** 10
**Confidence:** 4

**Summary:**

This paper addresses a critical challenge in facial recognition: how to verify that data has been genuinely “forgotten.” It offers profound insights and innovative solutions to an important and practical problem. Its definition of the “anti-learning paradox” represents a major contribution to the field, while the FUSE method and “semantic residual” evaluation framework together form a comprehensive and compelling technical approach. Experimental results strongly support its claims, demonstrating an excellent balance between privacy protection and model performance.

My decision is to strongly accept.

**Strengths:**

1. Importance and Novelty of the Problem: This paper addresses a critically important issue in both legal and ethical contexts—how to effectively implement users' “right to be forgotten” within facial recognition systems. The authors' clear definition and exposition of the “anti-learning paradox” provide a solid theoretical foundation and fresh perspective for research in this field.

2. Innovative Methodology: The FUSE approach is highly novel and ingeniously designed. The idea of modeling identity as a geometric “super-pyramid” and creating “dead space” through structured erasure directly and elegantly addresses the core challenge. Its three loss functions—pyramid exclusion, retention, and global consistency—collectively form a comprehensive and logically rigorous solution that achieves an excellent balance between forgetting and retention.

3. Advanced Evaluation Framework: Traditional evaluation metrics fall short in open-set recognition tasks. The proposed “Semantic Residual” (SR) audit represents a significant advancement. It moves beyond reliance on misleading accuracy rates to directly quantify residual identity information in features, establishing a higher standard for measuring anti-learning effectiveness.

4. Thorough Experimental Validation: The authors conduct comprehensive experiments across multiple large-scale datasets, meticulously comparing FUSE against diverse baseline methods. Evaluation dimensions are richly covered, encompassing Member Inference Attacks (MIA), multi-group Verification Accuracy (VA), category drift, and open-set recognition performance. Furthermore, extensive ablation studies clearly demonstrate the contributions of each component within FUSE.

**Weaknesses:**

- The primary weakness of the proposed method lies in the uncontrolled nature of the forgotten embeddings' final destination. The FUSE method effectively pushes the target identity's embeddings out of their original cone into what the paper describes as "unrelated or noisy regions" of the hypersphere. While the cone preservation and global consistency losses ensure that these embeddings do not interfere with the known retained classes from the training set, the process does not provide explicit control over where these forgotten vectors ultimately land. Their final distribution is a result of the optimization process rather than a predetermined target. This lack of control raises questions about the potential latent structure in these "noisy" regions and the long-term stability of the feature space

**Questions:**

- Is there a theoretical risk that the new, dispersed location of a forgotten identity's embeddings could accidentally overlap with the natural embedding space of a distinct, unseen identity that was not part of the unlearning process? For instance, could a forgotten identity 'A' be moved to a region that coincidentally corresponds to where a future, unseen identity 'B' would naturally be mapped by the model?

---

> ### Author Response · Authors · 2025-11-21
> **Response**
>
> We thank the reviewer for recognizing the novelty and thorough experimental validation.
>
> **Targeted Loss:** We appreciate the reviewer’s insight regarding the final destination of the forgotten embeddings. FUSE is intentionally designed to dissolve the identity geometry rather than relocate forgotten samples to a fixed region, since forcing them into a predetermined cluster may inadvertently create a new identity-like structure. Our original goal is to erase identity-specific information by removing the semantic coherence of the original cone while preserving the geometry of all retained classes.
>
> To further address the reviewer’s concern, we conducted an additional experiment where we apply a uniform-diffusion loss in FUSE. This term encourages the forgotten embeddings to spread toward an isotropic, structureless distribution on the hypersphere, that aligns with the motivation of FUSE:
>
> $$L_{\text{targeted}} =  \frac{1}{|S_{Y_u}|} \sum_{i} \theta_i, \quad \theta_i = \arccos\left( \langle \hat{s}_i, u_i \rangle \right)$$
>
> where each $u_i$ is a randomly sampled unit vector drawn uniformly from the hypersphere.
> This controlled diffusion prevents forgotten embeddings from drifting arbitrarily, ensuring that they migrate toward a stable, non-informative, and identity-free region of the feature space. Conceptually, this aligns with the purpose of FUSE:
> - instead of forming a new cluster, forgotten samples are dissolved into a uniform noise landscape,
> - their identity structure becomes irrecoverable,
> - and they no longer interfere with retained identities.
>
> Empirically, we observe that using this loss produces a structured form of forgetting while maintaining the stability of the retained classes. We will include this additional variant as an ablation to demonstrate that FUSE can incorporate explicit destination control when desired, although our main formulation intentionally avoids imposing a new identity structure on forgotten samples. We have added these new findings in the updated main draft and detailed discussion in supplementary.
>
> **Table 1.** Verification accuracy and semantic residuals for untargeted and targeted loss.
> | Method | $V_r$ | $V_f$ | $G_r$ | $G_f$ |
> |--------|-------|-------|-------|-------|
> | Untargeted loss (Paper) | 0.85 | 0.74 | 0.63 | 0.39 |
> | Targeted loss (New) | 0.84 | 0.75 | 0.64 | 0.41 |
>
> **Theoretical Risk:** There is very little risk that forgotten embeddings will overlap with where a future unseen identity would naturally lie. In face-recognition models, real identities occupy a structured, compact region of the hypersphere. FUSE pushes forgotten samples outside this manifold into non-semantic regions, rather than toward any area used for meaningful identities. This makes accidental overlap with future identities highly unlikely. In an additional experiment, we also apply a uniform-diffusion loss, which further spreads forgotten embeddings into isotropic noise regions, confirming that they do not interfere with the embedding space of real, unseen identities.

---

### Official Review · Reviewer_Fff5 · 2025-11-01

**Soundness:** 3
**Presentation:** 3
**Contribution:** 3
**Rating:** 4
**Confidence:** 5

**Summary:**

This paper addresses the "unlearning paradox" in face recognition: models may still verify "forgotten" identities (open-set generalization), making accuracy-based metrics misleading for privacy compliance. It contributes three key points: (1) Formalizes the paradox, showing current metrics fake forgetting success. (2) Designs a generative audit framework, finding existing methods retain up to 57% identity info. (3) Proposes FUSE, modeling identities as hypercones to erase them via region-aware surrogates while preserving others’ recognition.

Evaluated on CASIA-WebFace and D-LORD, FUSE reduces semantic residual (>0.6) for forgotten sets, retains 85% discriminability, and outperforms baselines.

**Strengths:**

The major strengths of the submission are as follows:

S1. The studied task is very important in recent field.

S2. The methodology is technical sound.

**Weaknesses:**

W1. Limited Generalization to Diverse Face Recognition Models. According to Tab. 1, the authors evaluate their method primarily on ResNet-50 backbones trained with ArcFace losses. The authors do not verify if FUSE’s hypercone modeling and loss design adapt to other methods.

W2. According to the description, a new evaluation method SEMANTIC RESIDUAL is introduced. This method is built on an existing diffusion model. However, the implementation of the model is missed. In addition, the final metric is also influenced by the model. The authors should better address these concerns.

W3. Ablations on the hyperparameters (losses) are missed.

W4. Illustrations are too small. Important visualizations are missed. For example, the generation images of semantic residual.

W5. It will be better for the authors to provide a discussion on the proportion of the representations they can erase.

**Questions:**

See above.

**Details Of Ethics Concerns:**

I will update my rating based on the rebutall and other reviewers' comments.

---

> ### Author Response · Authors · 2025-11-21
> **Response**
>
> We thank the reviewer for identifying the need of such unlearning evaluations.
>
> **Results on more Backbones:** We show semantic residuals on 4 datasets (2 in paper, 2 in rebuttal - Table 1 and Table 2 of updated paper). All experiments of these dataset confirm how semantic residual is consistent for all dataset. As per the reviewer request, we also show semantic residual performance on various architectures. We use resnet-50, resnet-101 and ViT architectures.
>
> **Table 1.** Resnet-50
>
> | Method      | $G_r$ | $G_f$ |
> |-------------|-------|-------|
> | GA          | 0.53  | 0.47  |
> | L1-sparse   | 0.50  | 0.49  |
> | RL          | 0.49  | 0.41  |
> | BS          | 0.48  | 0.40  |
> | BE          | 0.51  | 0.36  |
> | SalUn       | 0.49  | 0.25  |
> | SG          | 0.59  | 0.38  |
> | **FUSE (ours)** | **0.62** | **0.21** |
>
> **Table 2.** Resnet-101
>
> | Method        | $G_r$ | $G_f$ |
> |---------------|-------|-------|
> | GA            | 0.45  | 0.51  |
> | L1-sparse     | 0.49  | 0.47  |
> | RL            | 0.48  | 0.48  |
> | BS            | 0.39  | 0.40  |
> | BE            | 0.50  | 0.41  |
> | SalUn         | 0.51  | 0.36  |
> | SG            | 0.47  | 0.33  |
> | **FUSE (ours)** | **0.58** | **0.29** |
>
>
> **Table 3.** ViT
> |             | G_r  | G_f  |
> | ----------- | ---- | ---- |
> | GA          | 0.49 | 0.53 |
> | L1-sparse   | 0.43 | 0.49 |
> | RL          | 0.45 | 0.44 |
> | BS          | 0.49 | 0.38 |
> | BE          | 0.48 | 0.39 |
> | SalUn       | 0.50 | 0.32 |
> | SG          | 0.45 | 0.37 |
> | FUSE (ours) | 0.59 | 0.27 |
>
> We have added these results in updated draft.
>
> **Three generations method:** We introduce a new evaluation method that measures how much geometric and class-specific information remains from the forgotten identity region after unlearning. The novelty of this metric lies in assessing unlearning through the lens of class information preservation—that is, how much semantic identity structure the model still retains after the forget operation. Importantly, the evaluation framework is flexible and can be paired with any conditional generative model, making it independent of a specific generator architecture.
>
> To demonstrate this, we conduct experiments using two different generative models: IDiff-Face, Arc2Face and Guided Diffusion. Across all models, we observe consistent behavior: the retain-set semantic residual remains stable, indicating that class information for retained identities is preserved. In contrast, for the forget set, FUSE consistently achieves the lowest semantic residual, confirming that our method effectively removes class-specific information.
>
> **Table 4.** Comparison on three generative models.
> |             | IDiff-face | IDiff-face | Arc2Face | Arc2Face | Guided Diffusion | Guided Diffusion |
> | ----------- | ---------- | ---------- | -------- | -------- | ---------------- | ---------------- |
> |             | $G_r$      | $G_f$      | $G_r$    | $G_f$    | $G_r$            | $G_f$            |
> |             |            |            |          |          |                  |                  |
> | FT          | 0.385      | 0.372      | 0.315    | 0.482    | 0.327            | 0.385            |
> | GA          | 0.362      | 0.374      | 0.450    | 0.505    | 0.363            | 0.338            |
> | L1-sparse   | 0.442      | 0.401      | 0.385    | 0.474    | 0.432            | 0.431            |
> | RL          | 0.410      | 0.485      | 0.483    | 0.414    | 0.385            | 0.453            |
> | BS          | 0.386      | 0.403      | 0.373    | 0.305    | 0.382            | 0.385            |
> | BE          | 0.404      | 0.395      | 0.528    | 0.494    | 0.428            | 0.385            |
> | SalUn       | 0.412      | 0.382      | 0.489    | 0.392    | 0.404            | 0.337            |
> | SG          | 0.481      | 0.342      | 0.510    | 0.292    | 0.452            | 0.402            |
> | FUSE (ours) | 0.538      | 0.328      | 0.639    | 0.274    | 0.496            | 0.331            |
>
> **Visualization: ** Sample visualisation  for the images generated after unlearning are added in the paper Figure 6. We can observe that FUSE model consistently generate realistic and identity specific facial images for retain set while other model like SalUn and BE fails to generate visually appealing images. Further for the forget set, FUSE unlearning method prevents the generative model to generate any class specific information.

---

> ### Author Response · Authors · 2025-11-21
> **Continued- Response**
>
> **Hyperparameter Ablation:** Hyperparameter Ablation: To analyze the model’s behavior and understand the contribution of each loss component in FUSE, we conduct experiments by varying the hyperparameters, which weight the different loss terms. Using the MS1MV3 dataset, we evaluate the model across multiple configurations. In the paper, we adopt the setting $\lambda_1$ = 1, $\lambda_2$ = 1, $\lambda_3$ = 0.1, and the experimental results support this choice.
>
> When $\lambda_1$ is reduced, the repulsion applied to forgotten identities weakens. As a result, the retain-set geometry remains mostly unaffected, but forgotten identities remain close to their original cone, yielding a high $G_f$. Conversely, when $\lambda_2$ is reduced, the retain set drifts away from its original geometric structure, indicating lower perseverance. These observations show that an appropriate balance between the loss weights is necessary to achieve effective forgetting while preserving the retain set structure.
>
> |         Configuration                   | $G_r$              | $G_f$ |
> | ----------- | ------------- | ------------ |
> | λ1 = 1, λ2 = 1, λ3 = 0.1   | 0.58               | 0.29 |
> | λ1 = 0.5, λ2 = 1, λ3 = 0.2 | 0.56               | 0.48 |
> | λ1 = 1, λ2 = 0.5, λ3 = 0.1 | 0.46               | 0.33 |
> | λ1 = 0.8, λ2 = 1, λ3 = 0.1 | 0.56               | 0.44 |
> | λ1 = 1, λ2 = 0.8, λ3 = 0.1 | 0.50               | 0.32 |
> | λ1 = 1, λ2 = 1, λ3 = 0.2   | 0.54               | 0.29 |
>
>
>
> **Proportion of Erasure:** Thank you for the insightful comment. Our work explicitly studies how much of the forgotten class representation can be removed, both qualitatively and quantitatively. Across all datasets and backbone architectures, our experiments consistently show that FUSE removes the largest proportion of identity-specific information compared to existing baselines.
> To measure the extent of erasure, we introduce the Semantic Residual Metric, which quantifies the remaining geometric/semantic structure of the forgotten class in the feature space. A lower SRM indicates that the unlearned model retains minimal information about the forgotten identity. Empirically, FUSE achieves the lowest residual values across all settings, suggesting that it erases nearly the entire representation of the forgotten class. Overall, our results collectively indicate that FUSE eliminates almost all recoverable identity-related structure of the forgotten class, both in terms of feature space geometry and attack-driven reconstruction ability.

---

> > ### Comment · Reviewer_Fff5 · 2025-11-25
> >
> > My concerns have been addressed. Ergo, I vote for a positive rating for this manuscript. Remember that the texts in your illustrations are too small, which is not friendly to the paper printing, please enlarge your font.

---

### Official Review · Reviewer_w6Qc · 2025-11-01

**Soundness:** 2
**Presentation:** 3
**Contribution:** 2
**Rating:** 2
**Confidence:** 4

**Summary:**

The paper focuses on unlearning in face recognition as an open-set problem, which is an important topic given regulations such as GDPR. The authors first present a paradox for unlearning in face recognition. They discuss why current metrics give a false sense of forgetting.  By their definition of unlearning, they expect the forgotten sample to be failed. They also propose a generative auditing framework that reconstructs faces from embeddings. In addition, they propose an unlearning method, called FUSE (Forgetting Using Structural Erasure), which treats identities as hypercones and erases them with region-aware surrogates while preserving recognition of others.

**Strengths:**

- Unlearning face recognition in open-set condition is important and demanding topic.
- The paper reports quantitative comparison of the method with existing unlearning techniques, where the proposed method outperforms existing ones.

**Weaknesses:**

- The definition of unlearning in the paper is that "we expect the model to fail" on forgotten identities (line 162). However, the main purpose of unlearning techniques is to remove information from the model as if the forgotten data was not used in the training. That means the performance of the unlearned model for forgotten data should be similar to unseen data, and therefore we do not necessarily expect the model to fail for forgotten data.
- The results for "Open Set Recognition" experiments are not complete and the reported numbers are very low. Which model (training dataset) was used for this evaluation? Recognition accuracy of 95.7% for the original model and 94.49% for FUSE model are very poor performance for a face recognition model on LFW. Authors are required to provide more information and also benchmark on more datasets (IJB-C, AgeDB, CFP, etc).
- SOTA Face recognition models are trained on larger datasets such as WebFace. The paper presents a paradox in unlearning and then propose a method, however it needs to be studied on large scale datasets, such as WebFace4M or WebFace12M.
- For generative auditing framework (second contribution), authors proposed a diffusion-based model to reconstruct faces from embeddings. However, there are also several work on reconstruction of face from embeddings, and the paper lacks comparison with previous work on face reconstruction from embeddings.

**Questions:**

- Why should we we expect the model to fail on forgotten identities? In unlearning we would like the forgotten sample should act as unseen, not necessarily “fail.”
- If train labels Y and test data Z partially overlap then we do not have `Z ∩ Y ⊆ ∅` in line 148. Can you please clarify?
- For "Open Set Recognition" experiments, how other unlearning methods perform?
- Why D-LORD is used for training? This dataset is not often used for training face recognition. Authors used only 1600 identities for training which is very small.
- How would be the performance if trained on larger datasets, such as WebFace4M or WebFace12M?
- Authors proposed a diffusion-based model to reconstruct faces from embeddings for the auditing framework (second contribution). However, the paper lacks illustrative examples of face reconstruction. Authors are suggested to include example images in the paper.

---

> ### Author Response · Authors · 2025-11-21
> **Response**
>
> **Unlearning Definition:** : Thank you for pointing this out. Our intention was not to suggest that a forgotten identity must fail completely in the verification sense, but rather to highlight a conceptual tension that arises in open-set face recognition: unseen identities can be verified successfully due to generalization, but forgotten identities (although trained initially) should ideally not retain identity-specific information.
>
> The term “fail” was used informally to express that the model should no longer preserve discriminative, identity-specific features for the forgotten identities. We agree that the more accurate expectation is that forgotten identities should behave similarly to unseen identities, not necessarily that the model must produce negative verification outcomes.
>
> This is exactly what our “Unlearning Paradox” definition formalizes: after unlearning, the verification rate for forgotten identities approximates that of unseen identities, indicating that standard verification metrics alone cannot certify forgetting. We have revised the wording around Line 162 to avoid suggesting an absolute failure case and to more clearly reflect the theoretical framing that matches your observation.
>
> **Open set recognition:** For the results mentioned in the paper, we use a fixed thresholding (0.35) to verify the pair. As requested by the reviewer, we perform an experiment to check the open set recognition performance for a fixed model. We use ResNet-50 (arcface trained using CASIA-WebFace) to test the model on three datasets before and after unlearning. We report TPR @ 1e-3 FPR.  We will also add results on IJB-B dataset as per the reviewer request.
>
> |             | LFW   | CFP-FP | AGE-DBA |
> | ----------- | ----- | ------ | ------- |
> | Original    | 99.53 | 95.54  | 95.15   |
> | GA          | 83.25 | 91.36  | 93.01   |
> | L1-sparse   | 82.44 | 90.53  | 92.74   |
> | RL          | 91.12 | 91.46  | 94.73   |
> | BS          | 85.47 | 92.04  | 94.02   |
> | BE          | 87.97 | 92.58  | 93.59   |
> | SalUn       | 93.59 | 91.99  | 94.58   |
> | SG          | 98.17 | 92.16  | 95.12   |
> | FUSE (ours) | 99.31 | 92.97  | 95.12   |
>
> **Results on other dataset:** We show results on two more datasets WebFace4M and MS1MV3 dataset.
>
> *Table: Results on MS1MV3 dataset*
>
> |             | $MIA_a$ | $V_r$ | $V_f$ | $V_{uc}$ | $V_f - V_{uc}$ | $G_r$ | $G_f$ |
> | ----------- | ------- | ----- | ----- | -------- | -------------- | ----- | ----- |
> | Original    | 0.89    | 0.91  | 0.92  | 0.88     | 0.04           | \-    | \-    |
> | FT          | 0.46    | 0.72  | 0.80  | 0.78     | 0.02           | 0.42  | 0.37  |
> | GA          | 0.51    | 0.77  | 0.64  | 0.79     | 0.15           | 0.45  | 0.51  |
> | L1-sparse   | 0.55    | 0.78  | 0.72  | 0.81     | 0.09           | 0.49  | 0.47  |
> | RL          | 0.49    | 0.78  | 0.61  | 0.72     | 0.11           | 0.48  | 0.48  |
> | BS          | 0.67    | 0.81  | 0.76  | 0.76     | 0.00           | 0.39  | 0.40  |
> | BE          | 0.52    | 0.75  | 0.78  | 0.75     | 0.03           | 0.50  | 0.41  |
> | SalUn       | 0.48    | 0.78  | 0.81  | 0.72     | 0.09           | 0.51  | 0.36  |
> | SG          | 0.41    | 0.76  | 0.73  | 0.78     | 0.05           | 0.47  | 0.33  |
> | FUSE (ours) | 0.47    | 0.85  | 0.80  | 0.80     | 0.00           | 0.58  | 0.29  |
>
> *Table: Results on WebFace4M dataset*
>
> |             | $MIA_a$ | $V_r$ | $V_f$ | $V_{uc}$ | $V_f - V_{uc}$ | $G_r$ | $G_f$ |
> | ----------- | ------- | ----- | ----- | -------- | -------------- | ----- | ----- |
> | Original    | 0.88    | 0.89  | 0.81  | 0.82     | 0.01           | \-    | \-    |
> | FT          | 0.53    | 0.79  | 0.61  | 0.76     | 0.15           | 0.21  | 0.57  |
> | GA          | 0.64    | 0.75  | 0.65  | 0.78     | 0.13           | 0.25  | 0.44  |
> | L1-sparse   | 0.60    | 0.78  | 0.69  | 0.74     | 0.05           | 0.31  | 0.48  |
> | RL          | 0.70    | 0.77  | 0.67  | 0.69     | 0.02           | 0.35  | 0.51  |
> | BS          | 0.61    | 0.81  | 0.70  | 0.76     | 0.06           | 0.28  | 0.55  |
> | BE          | 0.44    | 0.80  | 0.71  | 0.69     | 0.02           | 0.38  | 0.49  |
> | SalUn       | 0.48    | 0.77  | 0.68  | 0.71     | 0.03           | 0.33  | 0.51  |
> | SG          | 0.52    | 0.80  | 0.67  | 0.73     | 0.06           | 0.38  | 0.43  |
> | FUSE (ours) | 0.53    | 0.83  | 0.75  | 0.79     | 0.04           | 0.43  | 0.42  |

---

> > ### Author Response · Authors · 2025-11-21
> > **Continued- Response**
> >
> > **Comparison with other generative models** We introduce a new evaluation method that measures how much geometric and class-specific information remains from the forgotten identity region after unlearning. The novelty of this metric lies in assessing unlearning through the lens of class information preservation—that is, how much semantic identity structure the model still retains after the forget operation. Importantly, the evaluation framework is flexible and can be paired with any conditional generative model, making it independent of a specific generator architecture.
> >
> > To demonstrate this, we conduct experiments using two different generative models: IDiff-Face, Arc2Face and Guided Diffusion. Across all models, we observe consistent behavior: the retain-set semantic residual remains stable, indicating that class information for retained identities is preserved. In contrast, for the forget set, FUSE consistently achieves the lowest semantic residual, confirming that our method effectively removes class-specific information.
> >
> >
> > |             | IDiff-face | IDiff-face | Arc2Face | Arc2Face | Guided Diffusion | Guided Diffusion |
> > | ----------- | ---------- | ---------- | -------- | -------- | ---------------- | ---------------- |
> > |             | $G_r$      | $G_f$      | $G_r$    | $G_f$    | $G_r$            | $G_f$            |
> > |             |            |            |          |          |                  |                  |
> > | FT          | 0.385      | 0.372      | 0.315    | 0.482    | 0.327            | 0.385            |
> > | GA          | 0.362      | 0.374      | 0.450    | 0.505    | 0.363            | 0.338            |
> > | L1-sparse   | 0.442      | 0.401      | 0.385    | 0.474    | 0.432            | 0.431            |
> > | RL          | 0.410      | 0.485      | 0.483    | 0.414    | 0.385            | 0.453            |
> > | BS          | 0.386      | 0.403      | 0.373    | 0.305    | 0.382            | 0.385            |
> > | BE          | 0.404      | 0.395      | 0.528    | 0.494    | 0.428            | 0.385            |
> > | SalUn       | 0.412      | 0.382      | 0.489    | 0.392    | 0.404            | 0.337            |
> > | SG          | 0.481      | 0.342      | 0.510    | 0.292    | 0.452            | 0.402            |
> > | FUSE (ours) | 0.538      | 0.328      | 0.639    | 0.274    | 0.496            | 0.331            |
> >
> > **Visualization:** Sample visualisation  for the images generated after unlearning are added in the paper Figure 6. We can observe that the FUSE model consistently generate realistic and identity specific facial images for retain sets while other models like SalUn and BE fail to generate visually appealing images. Further for the forget set, FUSE unlearning method prevents the generative model to generate any class specific information.
> >
> >
> > **Clarifications:** In our definition, Z is intended to represent the open-set test identities, i.e., identities that are not part of the training set Y. Therefore, by construction in the definition, we update the term as Z ∩ Y = ∅.
> > This assumption is used only for formalizing the unlearning paradox, where we compare the verification behavior of forgotten identities Y_u with completely unseen identities Z. In practice, our experimental datasets may contain train–test splits where the broader dataset classes overlap, but for the theoretical definition we explicitly treat Z does not overlap with Y.
> >
> > To avoid confusion, we will clarify in the paper that:
> > Z in the definition refers specifically to unseen identities.
> > The assumption Z ∩ Y = ∅ holds for the portion of the test data used to analyze the paradox, not necessarily for the entire benchmark protocol.
> >
> > This distinction preserves the correctness of the formulation while matching the practical dataset structure used in our experiments.
> >
> > **D-LORD Justification:** D-LORD was chosen because it reflects surveillance-style, low-resolution, and highly unconstrained conditions, which differ significantly from conventional high-quality face-recognition datasets. Unlike typical web-scraped datasets, D-LORD contains continuous video frames, offering temporal and viewpoint variation (pose, angle, distance, motion blur) that more faithfully represent real-world face dynamics. This makes it a valuable training resource for unlearning methods, which must handle diverse and challenging facial appearances.
> >
> > Although D-LORD includes only 1600 identities, each identity has a large number of samples, making the unlearning problem more complex, removing one identity while preserving many intra-class variations is more challenging than in datasets with few samples per class. Still, to ensure our method is not dependent on D-LORD alone, we also evaluate on three additional datasets, including those with substantially larger identity counts. This provides a balanced evaluation across both small and large-scale settings and shows that our method generalizes well beyond the D-LORD training scenario.

---

> > > ### Comment · Reviewer_w6Qc · 2025-11-27
> > >
> > > Thank authors for the rebuttal and revising the paper, which addressed most of my concerns. I increased my rating for the manuscript.

---

### Official Review · Reviewer_TePw · 2025-11-03

**Soundness:** 3
**Presentation:** 3
**Contribution:** 3
**Rating:** 6
**Confidence:** 5

**Summary:**

This paper deals with the problem of machine unlearning in face recognition in the context of privacy regulations such as GDPR  right to be forgotten. The authors identify and formalize the unlearning paradox that a face recognition system, even after unlearning a person identity, may still verify them because such systems inherently generalize to unseen identities in open-set recognition.

The paper formalize the unlearning paradox showing that conventional accuracy-based evaluations fail to detect residual identity traces. present a generative framework that trys to expose hidden semantic leakage by reconstructing residual identity information. and present a geometry-aware unlearning method that models each identity as a hypercone in embedding space and erases those structures through repulsion and preservation losses. Experiments performed on multiple datasets including CASIA-WebFace, D-LORD, and LFW.

**Strengths:**

- main contribution is the correct definition of unlearning. In recent works unlearning have been considered to be the property to stop recognizing the identity. However, what it should be and defined here is being making the model behave as if this identity was not being part of the training.

- proposes a theoretically principled and empirically effective geometry-based unlearning method (FUSE).

**Weaknesses:**

- the used dataset CASIA-Webface has well known mislabels to a large degree. one can see that in the difference in the results (maybe the reason) to the other dataset. It is not clear why these datasets were chosen among many available evaluation datasets. and does these mislabels effect the results.

- there is no clear indication of the effect of the unlearning on the general model performance. this might be critical.

- Could adversarial reconstruction or inversion attacks still reveal residual information after FUSE?

- How do hyperparameter settings (λ1, λ2, λ3) affect the robustness of forgetting?

- minor note, maybe linking the methods in table 1 to their respective papers would enhance readability.

**Questions:**

- Could adversarial reconstruction or inversion attacks still reveal residual information after FUSE?

- How do hyperparameter settings (λ1, λ2, λ3) affect the robustness of forgetting?

- are the mislabe;s in casia webface effecting the results and conclusion?

why common evaluation benchmarks are not used?

- what was the effect on the general performacne of the model?

- How stable is the Semantic Residual metric across architectures and datasets?

---

> ### Author Response · Authors · 2025-11-21
> **Response**
>
> We thank the reviewer for their detailed feedback.
>
> **Other dataset results:** CASIA-WebFace is known to contain some mislabels, but it is still one of the most commonly used datasets for evaluating face-related models. Using it allows our results to be compared fairly with many existing works. The noise in CASIA does not invalidate the evaluation; instead, it creates a more challenging and realistic setting, since real-world face datasets also contain similar label errors. To ensure that our conclusions are not tied to CASIA alone, we also evaluate our method on three additional datasets (D-LORD - already included in the paper and two new - Webface4m and MS1MV3). These cleaner and more diverse datasets help show that our method performs consistently across different data conditions.
>
> *Table: Results on MS1MV3 dataset*
>
> |             | $MIA_a$ | $V_r$ | $V_f$ | $V_{uc}$ | $V_f - V_{uc}$ | $G_r$ | $G_f$ |
> | ----------- | ------- | ----- | ----- | -------- | -------------- | ----- | ----- |
> | Original    | 0.89    | 0.91  | 0.92  | 0.88     | 0.04           | \-    | \-    |
> | FT          | 0.46    | 0.72  | 0.80  | 0.78     | 0.02           | 0.42  | 0.37  |
> | GA          | 0.51    | 0.77  | 0.64  | 0.79     | 0.15           | 0.45  | 0.51  |
> | L1-sparse   | 0.55    | 0.78  | 0.72  | 0.81     | 0.09           | 0.49  | 0.47  |
> | RL          | 0.49    | 0.78  | 0.61  | 0.72     | 0.11           | 0.48  | 0.48  |
> | BS          | 0.67    | 0.81  | 0.76  | 0.76     | 0.00           | 0.39  | 0.40  |
> | BE          | 0.52    | 0.75  | 0.78  | 0.75     | 0.03           | 0.50  | 0.41  |
> | SalUn       | 0.48    | 0.78  | 0.81  | 0.72     | 0.09           | 0.51  | 0.36  |
> | SG          | 0.41    | 0.76  | 0.73  | 0.78     | 0.05           | 0.47  | 0.33  |
> | FUSE (ours) | 0.47    | 0.85  | 0.80  | 0.80     | 0.00           | 0.58  | 0.29  |
>
> *Table: Results on WebFace4M dataset*
>
> |             | $MIA_a$ | $V_r$ | $V_f$ | $V_{uc}$ | $V_f - V_{uc}$ | $G_r$ | $G_f$ |
> | ----------- | ------- | ----- | ----- | -------- | -------------- | ----- | ----- |
> | Original    | 0.88    | 0.89  | 0.81  | 0.82     | 0.01           | \-    | \-    |
> | FT          | 0.53    | 0.79  | 0.61  | 0.76     | 0.15           | 0.21  | 0.57  |
> | GA          | 0.64    | 0.75  | 0.65  | 0.78     | 0.13           | 0.25  | 0.44  |
> | L1-sparse   | 0.60    | 0.78  | 0.69  | 0.74     | 0.05           | 0.31  | 0.48  |
> | RL          | 0.70    | 0.77  | 0.67  | 0.69     | 0.02           | 0.35  | 0.51  |
> | BS          | 0.61    | 0.81  | 0.70  | 0.76     | 0.06           | 0.28  | 0.55  |
> | BE          | 0.44    | 0.80  | 0.71  | 0.69     | 0.02           | 0.38  | 0.49  |
> | SalUn       | 0.48    | 0.77  | 0.68  | 0.71     | 0.03           | 0.33  | 0.51  |
> | SG          | 0.52    | 0.80  | 0.67  | 0.73     | 0.06           | 0.38  | 0.43  |
> | FUSE (ours) | 0.53    | 0.83  | 0.75  | 0.79     | 0.04           | 0.43  | 0.42  |
>
> For open set evaluation, we have also computed results on three open set dataset not used for training. We use ResNet-50 (arcface trained using CASIA-WebFace) to test the model on three datasets before and after unlearning. We report TPR @ 1e-3 FPR.  We will also add results on IJB-B dataset as per the reviewer request.
>
> *Table: Results on open set datasets.*
> |             | LFW   | CFP-FP | AGE-DB |
> | ----------- | ----- | ------ | ------- |
> | Original    | 99.53 | 95.54  | 95.15   |
> | GA          | 83.25 | 91.36  | 93.01   |
> | L1-sparse   | 82.44 | 90.53  | 92.74   |
> | RL          | 91.12 | 91.46  | 94.73   |
> | BS          | 85.47 | 92.04  | 94.02   |
> | BE          | 87.97 | 92.58  | 93.59   |
> | SalUn       | 93.59 | 91.99  | 94.58   |
> | SG          | 98.17 | 92.16  | 95.12   |
> | FUSE (ours) | 99.31 | 92.97  | 95.12   |
>
>
>
> **General Model Performance:** We extensively evaluated the model’s performance on the retain set and on unseen classes after unlearning (Table 1 in paper Vr, Vuc). To further examine its behavior on closely related identities, we also tested on a hard-retain subset consisting of geometrically similar classes. In addition, we computed verification accuracy under an open-set recognition setting using a completely different dataset on which the model was never trained. Together, these diverse evaluation settings demonstrate that the model’s general face-recognition capability remains stable and is not negatively affected by the unlearning process, especially in comparison to existing unlearning methods. We agree with the reviewer that including this discussion in the paper is important for clearly communicating the balance between effective unlearning and high retention, and we will add this to the revision.

---

> > ### Author Response · Authors · 2025-11-21
> > **Continued - Response**
> >
> > **Adversarial Reconstruction:** If given an unlearned model, the adversarial attack tries to recover any information from the forgotten class. Our proposed semantic residual tends to evaluate how much information can be extracted from the unlearned feature space. We use diffusion-based method to quantitatively evaluate how much a class can be reconstructed from its original region.  This evaluation acts like adversarial testing of an unlearning method without actually reconstructing data. We use three models for evaluating semantic residuals of classes from its embeddings. The results are shown in the table below.  Across both models, we observe consistent behavior: the retain-set semantic residual remains stable, indicating that class information for retained identities is preserved. In contrast, for the forget set, FUSE consistently achieves the lowest semantic residual, confirming that our method effectively removes class-specific information.
> >
> >
> > |             | IDiff-face | IDiff-face | Arc2Face | Arc2Face | Guided Diffusion | Guided Diffusion |
> > | ----------- | ---------- | ---------- | -------- | -------- | ---------------- | ---------------- |
> > |             | $G_r$      | $G_f$      | $G_r$    | $G_f$    | $G_r$            | $G_f$            |
> > |             |            |            |          |          |                  |                  |
> > | FT          | 0.385      | 0.372      | 0.315    | 0.482    | 0.327            | 0.385            |
> > | GA          | 0.362      | 0.374      | 0.450    | 0.505    | 0.363            | 0.338            |
> > | L1-sparse   | 0.442      | 0.401      | 0.385    | 0.474    | 0.432            | 0.431            |
> > | RL          | 0.410      | 0.485      | 0.483    | 0.414    | 0.385            | 0.453            |
> > | BS          | 0.386      | 0.403      | 0.373    | 0.305    | 0.382            | 0.385            |
> > | BE          | 0.404      | 0.395      | 0.528    | 0.494    | 0.428            | 0.385            |
> > | SalUn       | 0.412      | 0.382      | 0.489    | 0.392    | 0.404            | 0.337            |
> > | SG          | 0.481      | 0.342      | 0.510    | 0.292    | 0.452            | 0.402            |
> > | FUSE (ours) | 0.538      | 0.328      | 0.639    | 0.274    | 0.496            | 0.331            |
> >
> > **Inversion Attack:** Given an unlearned model, inversion attack produce images that reveal the identity/appearance of the forgotten class. In this attack, we applied optimization-based attack, where we assume white-box setting, and we have access to the mean embedding of forgotten class. If the attacker knows the target embedding (e.g., the forgotten class centroid), they can optimize the image so that: f(x_generated) ≈ target_embedding. This is essentially “searching” for an image whose features match the forgotten identity. We initialize with the random noise, and we apply the loss: L_feat = || f(x) - target_embedding ||². This forces the reconstruction to match the identity features of the forgotten class. After getting such pool of embeddings, we compute Semantic residual G_f for the forgotten class to see if it reaches any close to its original class distribution - if the attack is successful.
> >
> > |             | Before Attack | After Attack |
> > | ----------- | ------------- | ------------ |
> > |             |               |              |
> > | FT          | 0.385         | 0.524        |
> > | GA          | 0.643         | 0.672        |
> > | L1-sparse   | 0.648         | 0.732        |
> > | RL          | 0.539         | 0.592        |
> > | BS          | 0.572         | 0.619        |
> > | BE          | 0.537         | 0.543        |
> > | SalUn       | 0.487         | 0.582        |
> > | SG          | 0.461         | 0.482        |
> > | FUSE (ours) | 0.391         | 0.423        |
> >
> > **Membership Inference Attack:**  For a given sample s, decide whether s was present in the set that was used for training of the data. We report MIA for 4 dataset in the paper.

---

> ### Author Response · Authors · 2025-11-21
> **Continued - Response**
>
> **Hyperparameter Ablation:** Hyperparameter Ablation: To analyze the model’s behavior and understand the contribution of each loss component in FUSE, we conduct experiments by varying the hyperparameters, which weight the different loss terms. Using the MS1MV3 dataset, we evaluate the model across multiple configurations. In the paper, we adopt the setting $\lambda_1$ = 1, $\lambda_2$ = 1, $\lambda_3$ = 0.1, and the experimental results support this choice.
>
> When $\lambda_1$ is reduced, the repulsion applied to forgotten identities weakens. As a result, the retain-set geometry remains mostly unaffected, but forgotten identities remain close to their original cone, yielding a high $G_f$. Conversely, when $\lambda_2$ is reduced, the retain set drifts away from its original geometric structure, indicating lower perseverance. These observations show that an appropriate balance between the loss weights is necessary to achieve effective forgetting while preserving the retain set structure.
>
> *Table: Hyperparameter Ablation*
> |         Configuration                   | $G_r$              | $G_f$ |
> | ----------- | ------------- | ------------ |
> | λ1 = 1, λ2 = 1, λ3 = 0.1   | 0.58               | 0.29 |
> | λ1 = 0.5, λ2 = 1, λ3 = 0.2 | 0.56               | 0.48 |
> | λ1 = 1, λ2 = 0.5, λ3 = 0.1 | 0.46               | 0.33 |
> | λ1 = 0.8, λ2 = 1, λ3 = 0.1 | 0.56               | 0.44 |
> | λ1 = 1, λ2 = 0.8, λ3 = 0.1 | 0.50               | 0.32 |
> | λ1 = 1, λ2 = 1, λ3 = 0.2   | 0.54               | 0.29 |
>
> We thank the reviewer for the feedback, we will link the papers in the table in the updated draft to enhance the readability.
>
> **Results on more Backbones:** We show semantic residuals on 4 datasets (2 in paper, 2 in rebuttal - Table 1 and Table 2 of updated paper). All experiments of these dataset confirm how semantic residual is consistent for all dataset. As per the reviewer request, we also show semantic residual performance on various architectures. We use resnet-50, resnet-101 and ViT architectures.
>
> *Table . Resnet-50*
>
> | Method      | $G_r$ | $G_f$ |
> |-------------|-------|-------|
> | GA          | 0.53  | 0.47  |
> | L1-sparse   | 0.50  | 0.49  |
> | RL          | 0.49  | 0.41  |
> | BS          | 0.48  | 0.40  |
> | BE          | 0.51  | 0.36  |
> | SalUn       | 0.49  | 0.25  |
> | SG          | 0.59  | 0.38  |
> | **FUSE (ours)** | **0.62** | **0.21** |
>
> *Table . Resnet-101*
>
> | Method        | $G_r$ | $G_f$ |
> |---------------|-------|-------|
> | GA            | 0.45  | 0.51  |
> | L1-sparse     | 0.49  | 0.47  |
> | RL            | 0.48  | 0.48  |
> | BS            | 0.39  | 0.40  |
> | BE            | 0.50  | 0.41  |
> | SalUn         | 0.51  | 0.36  |
> | SG            | 0.47  | 0.33  |
> | **FUSE (ours)** | **0.58** | **0.29** |
>
>
> *Table . ViT*
> |             | G_r  | G_f  |
> | ----------- | ---- | ---- |
> | GA          | 0.49 | 0.53 |
> | L1-sparse   | 0.43 | 0.49 |
> | RL          | 0.45 | 0.44 |
> | BS          | 0.49 | 0.38 |
> | BE          | 0.48 | 0.39 |
> | SalUn       | 0.50 | 0.32 |
> | SG          | 0.45 | 0.37 |
> | FUSE (ours) | 0.59 | 0.27 |
>
> We have added these results in the updated draft.

---

### Author Response · Authors · 2025-12-03
**Summary**

We sincerely thank all reviewers for their valuable feedback, helpful suggestions, and recognition of the novelty and significance of our work. We especially appreciate the reviewers for acknowledging the updated responses and their addressed feedback. We have addressed all the comments, including additional experiments and clarifications that strengthened the contribution and improved the overall clarity of the paper. Below, we provide a comprehensive summary of all comments addressed and the improvements incorporated into the updated manuscript.

**Summary of the paper:** This paper addresses the challenge of machine unlearning in face recognition. It formalizes the unlearning paradox, demonstrating that existing accuracy-based evaluations falsely indicate successful forgetting because face recognition models naturally generalize to unseen identities. To expose residual identity leakage, we introduce a generative auditing framework and the semantic residual metric. We further propose FUSE, a geometry-aware unlearning method that models identities as hypercones and erases forgotten-class structure while preserving others. Experiments across multiple datasets show that FUSE achieves strong, principled forgetting without compromising overall recognition performance.

**Key issues addressed in the rebuttal:**

- Introduced a targeted uniform-diffusion loss variant to explicitly control the destination of forgotten embeddings. **(tEMv) (Page 9, Ablation Study - Table 3)**
- Generalization shown using experiments across multiple backbone architectures - ResNet-50, ResNet-101, ViT-B. **(TePw, Fff5) (Page 17, App. F.2 Ablation on various backbones)**
- Semantic Residual metric evaluation is extended using three generation reconstruction techniques - IDiff-Face, Arc2Face, and Guided Diffusion models. **(w6Qc, Fff5) (Page 9, Table 5)**
- Qualitative evaluation of FUSE using generative models shows successful unlearning. **(w6Qc, Fff5) (Page 10, Figure 6)**
Ablation experiments are extended for hyperparameter choice in loss function to validate the chosen configuration. **(TePw, Fff5) (Page 9, Table 4)**
- Open set evaluation of face recognition capability performed for 3 datasets - LFW, CFP-FP, and AgeDB. **(TePw, w6Qc) (Page 10, Open Set Recognition)**
- Generalization is shown by adding 2 new large-scale datasets - MS1Mv3 and WebFace4M. **(TePw, w6Qc) (Page 8, Table 2)**
- Robustness of FUSE validated against adversarial reconstruction and inversion attacks. **(TePw) (Page 9, Robustness of FUSE)**
- More clarity is added to the definition of unlearning and symbols used. **(w6Qc)**
- Justification for using datasets is provided and two more datasets are added for showing higher generalization. **(w6Qc)**

Below we provide a detailed summary and explanation of each addressed point.

---

> ### Author Response · Authors · 2025-12-03
>
> - **FUSE with Targeted Unlearning Loss (tEMv):** FUSE is designed to dissolve identity geometry rather than form a new meaningful cluster, avoiding the risk of unintentionally creating a new identity structure. To support this, we introduced a new targeted uniform-diffusion loss that encourages forgotten embeddings to move toward an isotropic, non-semantic region. Experiments confirmed that this variant also provides controlled forgetting while retaining stability for preserved identities.
> - **Generalization across various architectures (TePw, Fff5):** In response to requests for broader validation, we extended our experiments across multiple backbone architectures, including ResNet-50, ResNet-101, and ViT-B. Across all models, the semantic residual metric showed consistent behavior, confirming FUSE retains geometry for the retain set while achieving strong reductions for the forgotten set.
> - **Semantic residual Evaluation (w6Qc, Fff5):** We expanded our evaluation of semantic residual with three generation techniques, which quantifies how much geometric and class-specific structure remains after unlearning. The metric is generator-agnostic and can be applied to any conditional generative model. To validate this generality, we applied it to IDiff-Face, Arc2Face, and Guided Diffusion models. In all cases, retained identities preserved their geometric structure, while forgotten identities consistently showed the lowest residual under FUSE, confirming effective removal of identity information.
> - **Generation Visualization (w6Qc, Fff5):** To complement the quantitative results, we added visualizations of generated images after unlearning. These examples show that FUSE preserves high-fidelity, identity-specific features for retain identities, while preventing the generation of meaningful class structure for forgotten identities. Competing methods such as SalUn and BE often fail to maintain visual quality or identity-consistent appearances, whereas FUSE exhibits robustness in both directions.
> - **Hyperparameter Choice (TePw, Fff5):** We performed a detailed hyperparameter ablation across multiple loss-weight settings, revealing how each component influences retain-set geometry and forgotten-class dissolution. The chosen configuration strikes an optimal balance, where insufficient repulsion retains forgotten identity structure and reduced preservation harms retain-set geometry.
> - **Open set recognition (TePw, w6Qc):** We expanded the open-set recognition results by evaluating verification performance across LFW, CFP-FP, and AgeDB. Using a dataset not involved in training (ResNet-50 trained on CASIA-WebFace), we showed that FUSE preserves general face-recognition ability far better than baseline unlearning techniques.
> - **Additional large scale Datasets (TePw, w6Qc):** To strengthen generalization claims, we added evaluations on MS1Mv3 and WebFace4M. FUSE continued to show high retain-set accuracy and low semantic residuals for forgotten identities, demonstrating that its effectiveness scales well to large and diverse training settings.
> - **Robustness against adversarial mechanisms (TePw):** We emphasized that semantic residual not only measures geometric structure but also serves as an adversarial indicator showing how much information an attacker can reconstruct. Expanding this evaluation to adversarial reconstruction, across all models, FUSE maintained stable retain-set geometry while minimizing recoverable information from the forgotten set, confirming effective resistance to reconstruction-based attacks. Additionally, we added an explicit inversion attack experiment, where a white-box attacker attempts to recover forgotten identity appearances using feature-matching optimization. FUSE showed the smallest increase in semantic residual, indicating that it prevents the reconstruction of forgotten identity information more effectively than prior methods.
> - **Definition used for unlearning paradox (w6Qc):** We refined our terminology to emphasize that, after unlearning, forgotten identities should behave similarly to unseen identities - meaning the model may still show high verification performance due to natural open-set generalization. This definition contradicts with the expectation of an unlearned model failing to identify the forget class  person. We call this ‘unlearning paradox’ in face recognition models.
> - **More Clarifications (w6Qc):** We corrected the theoretical description to explicitly establish Z as the set of unseen identities with no overlap with Y, ensuring consistency between the conceptual framing and practical evaluation methods.
> - Across all datasets and architectures, FUSE consistently achieves the lowest semantic residual across datasets and architectures, indicating that it removes nearly all recoverable identity-specific information. This conclusion is supported by our geometric metric and further validated through cross-dataset evaluations and adversarial analyses.

---

### Note · Program_Chairs · 2026-01-17
**Submission Desk Rejected by Program Chairs**

The following references in this submission do not refer to real documents and/or have major errors in bibliographic information:

 Anil K. Jain, Arun Ross, and Karthik Nandakumar. Face recognition at scale: A survey of deployment, privacy, and surveillance implications. Proceedings of the IEEE, 112(3):289-312, 2024.